# A Probabilistic Basis for Low-Rank Matrix Learning

## Abstract

Low rank inference on matrices is widely conducted by optimizing a cost function augmented with a penalty proportional to the nuclear norm $\|\cdot\|_*$. However, despite the assortment of computational methods for such problems, there is a surprising lack of understanding of the underlying probability distributions being referred to. In this article, we study the distribution with density $f(X) \propto e^{-\lambda\|X\|_*}$, finding many of its fundamental attributes to be analytically tractable via differential geometry. We use these facts to design an improved MCMC algorithm for low rank Bayesian inference as well as to learn the penalty parameter $\lambda$, obviating the need for hyperparameter tuning when this is difficult or impossible. Finally, we deploy these to improve the accuracy and efficiency of low rank Bayesian matrix denoising and completion algorithms in numerical experiments.

## 1 Introduction

A physician searches for cancerous growths in her patient's lungs via medical imaging; a psychologist theorizes about common traits underlying answers to a personality test; a cybersecurity specialist inspects network traffic to detect anomalies. Each of these scenarios involves dealing with large matrices which cannot reasonably be modeled as an array of unrelated numbers: rather, each entry is a window into a common latent structure. In such situations, the *low rank* assumption can be quite fruitful. The bevy of applications boosted by this assumption and the rich mathematics underlying the associated methods have lead to an explosion of academic research in this area over the past several decades (see, e.g., the review articles Hu et al. (2021); Davenport & Romberg (2016); Nguyen et al. (2019)).

In the mathematics of machine learning, inference under these assumptions is expressed as a cost function $c$ over a matrix $X$ combined with a penalty term operationalizing the belief that the matrix $X$ is of low rank, most straightforwardly the rank function itself:

$$\min_{X \in \mathbb{R}^{m \times n}} c(X) + \lambda \mathrm{rank}(X). \tag{1}$$

However, the rank function is nonconvex and discontinuous, yielding a composite function which does not enjoy any continuous or convex structure which might be present in $c$. A popular alternative is thus to optimize a surrogate cost where the matrix rank function is replaced by the *nuclear norm* (also known as the Schatten 1-norm, trace norm, or Ky Fan norm (Fan, 1951)), which is given by the sum of the singular values of a matrix $\|X\|_* = \sum_{i=1}^{\min(n,m)} \sigma_i(X)$:

$$\min_{X \in \mathbb{R}^{m \times n}} c(X) + \lambda \|X\|_*. \tag{2}$$

As the convex envelope of the rank function Fazel et al. (2001); Fazel (2002), the nuclear norm is an effective surrogate which preserves any convexity in $c$, and as such, has seen considerable attention in terms of impactful applications Chen & Suter (2004); Ji et al. (2010); Bell et al. (2010); Candes et al. (2015); Shen et al. (2015); Nguyen et al. (2019), innovative methods Daubechies et al. (2004); Beck & Teboulle (2009); Liu & Vandenberghe (2010); Cai et al. (2010); Mazumder et al. (2010), and impressive theoretical guarantees Candes & Recht (2008); Candès & Tao (2010); Recht et al. (2010). Low rank structure is particularly useful for learning in data-limited scenarios or for discovering relationships in complex data.

There is a well known connection between penalized empirical loss minimization and *Maximum a Posteriori* (MAP) Bayesian inference which is achieved when the prior is given by the negated and exponentiated penalty. For instance, regression with the analog of the nuclear norm penalty for vectors, the $\ell_1$ penalty Taylor et al. (1979), is famously equivalent to MAP inference with a Laplace prior Tibshirani (1996). This connection helps build intuition about the behavior of $\ell_1$ penalized regression, suggests extensions through more sophisticated priors, and enables hierarchical Bayesian learning of penalty hyperparameters.

Despite the prevalence of the nuclear norm penalty in empirical risk minimization, there is little known about the associated probability distribution; namely, that with density proportional to $e^{-\lambda\|X\|_*}$. Indeed, we are not aware of any article in the academic literature giving even its normalizing constant. In addition to the foundational theoretical import of such a result, it is also essential for Bayesian hierarchical. For example, running an entire MCMC chain for a grid of $\lambda$ values is computationally prohibitive; this contrasts with the optimization case, which can take advantage of warm starts and fast convergence of quadratic methods. Knowing the normalizing constant allows for Bayesian estimation of this hyperparameter. The purpose of this article is to address these and other practical problems by establishing the basic theoretical properties of the distribution.

## 2 BACKGROUND

We review low rank inference via optimization and simulation, with a focus on previous sightings of this "nuclear norm distribution" (NND).

While the nuclear norm function is nonsmooth, it has a simple proximal operator (see Appendix A for an introduction to proximal operators), meaning that first order optimization methods can still be straightforwardly deployed on such problems. Denoising under isotropic Gaussian noise can be performed in closed form using this framework Bertero & Boccaci (1998), and an iteration of this can perform effective and scalable matrix completion via the ISTA algorithm Daubechies et al. (2004); Beck & Teboulle (2009). Segert (2024) showed that the nuclear norm retains its analytical tractability in the context of multivariate regression.

Though the nuclear norm has the advantage of being a convex relaxation of the rank function, the log determinant $\log(|X^\top X| + \epsilon)$ is also sometimes used as a rank-reducing penalizer Fazel (2002). Yang et al. (2018) studied the probabilistic foundations of this penalty, finding that its density could be written in terms of a Wishart mixture of normals.

Bayesian methods have also been proposed for low rank inference. Many authors prefer to perform low rank Bayesian inference not directly on $X$, but rather on a hypothesized factorization of $X = UV^\top$, often with $U, V$ containing iid normal entries. Several authors, particularly in Variational Bayesian inference, have previously noted a close connection between this "normal product" distribution and the nuclear norm distribution Srebro et al. (2004); Wipf (2016); Kim & Choi (2013), but the exact nature of the relationship appears unknown. Other distributions used for low rank inference include the singular matrix Gaussian (Yuchi et al., 2023). Though they use the nuclear norm estimate as an initialization for their sampler, they do not mention the implied distribution. See Alquier (2013) and Babacan et al. (2012) for a review of priors commonly used for low rank Bayesian inference. The nuclear norm distribution is also mentioned by Pereyra (2016), who uses it to illustrate a general purpose MCMC algorithm applicable to nonsmooth functions.

## 3 THEORETICAL ANALYSIS

In what follows, we will consider matrices of size $n \times m$ and denote $\mathrm{NND}_{n,m}(\lambda)$ the Nuclear Norm distribution, that is, the distribution over $\mathbb{R}^{n\times m}$ with density $\propto e^{-\lambda\|X\|_*}$. We also assume throughout, with no loss of generality, that $n \leq m$.

### 3.1 BASIC PROPERTIES

The very first preliminary property is whether $e^{-\lambda\|X\|_*}$ even corresponds to a valid probability density at all. Equivalently, is it true that $\int e^{-\lambda\|X\|_*} dX < \infty$? One easy way to see that this is true is to use the fact that all norms on a finite-dimensional Euclidean space are equivalent; thus

for example $\|X\|_* \geq C\|X\|_{L1}$ for some constant $C$ depending only on the dimensions of $X$, and therefore

$$\int e^{-\lambda\|X\|_*}dX \leq \int e^{-\lambda C\|X\|_{L1}}dX = \prod_{ij}\int e^{-\lambda C|X_{ij}|}dX_{ij} < \infty\,. \tag{3}$$

We will also determine the exact normalizing constant below, although the argument is more involved.

We can already deduce several simple but useful properties at this point.

**Proposition 3.1.** *The Nuclear Norm distribution is symmetric under $O(n) \times O(m)$ where $O(n)$ is the general orthogonal group on $\mathbb{R}^n$. That is, if $X \sim \mathrm{NND}(\lambda)$ and $U$ and $V$ are orthogonal matrices, then $UXV \sim \mathrm{NND}(\lambda)$.*

*Proof.* For fixed orthogonal $U, V$, it is easily seen that the mapping $X \mapsto UXV$ has Jacobian determinant $\pm 1$ Moreover, it is a basic fact about SVD that $\|X\|_* = \|UXV\|_*$ for any $X$. Thus, the proposition follows from the standard change-of-variables formula. $\square$

The following proposition is also simple to prove, but provides a useful characteristic "signature" of the distribution.

**Proposition 3.2.** *If $X \sim \mathrm{NND}(\lambda)$, then its nuclear norm $\|X\|_*$ is Gamma distributed with shape parameter $nm$ and rate parameter $\lambda$.*

*Proof.* Let $C(\lambda) := \int e^{-\lambda\|X\|_*}dX$ denote the normalizing constant (which we know is finite by the above discussion). By homogeneity of the nuclear norm, it is easy to see that $C(\lambda) = \lambda^{-nm}C(1)$ for any $\lambda > 0$.

Now if $t < \lambda$, then the moment generating function of $\|X\|_*$ is given by

$$m(t) := \mathbb{E}e^{t\|X\|_*} = C(\lambda)^{-1}\int e^{t\|X\|_*}e^{-\lambda\|X\|_*}dX = C(\lambda)^{-1}C(-t+\lambda) \tag{4}$$

$$= \lambda^{nm}(-t+\lambda)^{-nm} = (1-\frac{t}{\lambda})^{-nm}\,, \tag{5}$$

which is the MGF of the indicated Gamma distribution. $\square$

### 3.2 NORMALIZING CONSTANT

We now state the result for the exact normalizing constant. The proof involves using the Coarea Formula (see Appendix B for a review) to decompose the integral over the slices $\{X : \|X\|_* = t\}$, and is given in Appendix D .

**Proposition 3.3.** *The normalizing constant $C(\lambda) := \int e^{-\lambda\|X\|_*}dX$ is given by*

$$C(\lambda) = \frac{(nm-1)!}{\sqrt{\min(n,m)}}\mathrm{Vol}_{mn-1}(S^*(1))\lambda^{-nm}\,, \tag{6}$$

*where $S^*(1) := \{X : \|X\|_* = 1\}$, and $\mathrm{Vol}_{nm-1}$ is the $nm-1$-dimensional surface measure.*

### 3.3 EXACT STOCHASTIC REPRESENTATION

If $X$ is distributed according to the Nuclear norm Distribution, we seek an exact factorization for the distribution of the singular vectors and singular values of $X$. While this can be derived using standard results in random matrix theory, we will nonetheless find it useful to record the result for our setting:

**Proposition 3.4.** *If $X \sim \mathrm{NND}_{nm}(\lambda)$, and has singular value decomposition $X = U\mathrm{diag}(S)V^T$, then $U$, $V$ and $S$ are independent and have the following distributions:*

$$U \sim \mathrm{Unif}(O(n))\,; \qquad V \sim \mathrm{Unif}(S(m,n))\,,$$
$$P(S) \propto \mathbb{1}_{[s_i \geq 0\,\forall i]}e^{-\lambda\sum_i s_i}\prod_{i \leq n}s_i^{m-n}\prod_{i < j \leq n}|s_i^2 - s_j^2|\,,$$

*where $S(n, m)$ is the Stiefel manifold of all $m \times n$ orthonormal frames. Conversely, if $U$, $V$, and $S$ are independently distributed according to the above distributions, then the product $U \operatorname{diag}(S) V^T$ is distributed according to $\mathrm{NND}_{mn}$*

*Proof.* See for example Anderson (2010), Prop. 4.1.3. We also give an alternative argument in Appendix E, which is considerably simpler than the proof in the reference. □

A few remarks: first, because the distribution of $S$ is permutation invariant, the statement of the Proposition still holds if we impose an order constraint on the singular values (which allows us to drop the absolute values). Second, some care is required when obtaining the singular value decomposition from $X$ in the above theorem, since this is of course not uniquely determined. Although, with probability 1, there are a total of $2^{\min(n,m)} \min(n, m)!$ possibilities, corresponding to reordering the singular values, as well as sinultaneously multiplying the $ith$ column of $U$ and $V$ by $-1$. The random variables $U, S, V$ in the above theorem can be interpreted as uniformly random samples from the finite set of possible SVDs of $X$ (and undefined on the set of measure zero in which $X$ has repeated singular values) .

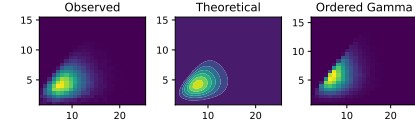

Figure 1: *Left:* Histogram of singular values of NND variates of size $7 \times 2$ and $\lambda = 1$. *Center:* Density of Theorem 3.4. *Right:* A histogram of ordered Gamma(7,1) variates.

At this point we can already use our results to clarify some previous claims made about the NND in the literature. For instance, Pereyra (2016) has commented on the NND as a matrix distribution extending the Laplace distribution "in which the singular values of [the matrix] are assigned exponential priors". This makes sense from an intuitive perspective, but already this idea was in trouble starting with Proposition 3.2: we cannot achieve a Gamma distribution with shape parameter $mn$ from only $m$ many exponential variates; we would need $mn$ many. However, this constraint on the sum can be satisfied if the singular values were ordered Gamma variates instead, each with a common rate parameter. Theorem 3.4 rules this out. The density there has substantive correlation between the singular values beyond their ordering, as illustrated in Figure 1.

Finally, although the decomposition in 3.4 is exact, the singular value density can be cumbersome to work with. This motivates our search for approximate stochastic representations that are easier to work with in practice.

### 3.4 APPROXIMATE STOCHASTIC REPRESENTATION: NORMAL PRODUCT DISTRIBUTION

We propose the following approximate stochastic representation of the NND. If $X_1$ and $X_2$ are $n \times n$ matrices with iid unit normal entries, then we propose to approximate $\mathrm{NND}_{nn}(1)$ with the distribution of $\frac{4}{3} X_1 X_2$. Note that this approximate distribution is very tractable both computationally (drawing prior samples is trivial and the conditional Gaussian prior facilitates posterior sampling) and analytically (we survey known results about its spectrum in Appendix G).

In the rest of this section, we will justify this approximation by a theoretical analysis. In Section 5.5, we will quantify the quality of this approximation by simulations.

Our approach is based on the variational characterization of the nuclear norm (Rennie, 2005):

$$\|X\|_* = \frac{1}{2} min_{U \in \mathbb{R}^{n \times d}, V \in \mathbb{R}^{d \times m}: UV = X} \|U\|_F^2 + \|V\|_F^2 \,; \qquad d = \min(n, m). \tag{7}$$

If we replace the minimum with a soft minimum, i.e. a log-integral-exp, then intuitively we expect:

$$e^{-\|X\|_*/\tau} \stackrel{?}{\approx} \int_{UV = X} e^{-\frac{1}{2\tau} \|U\|_F^2 - \frac{1}{2\tau} \|V\|_F^2} d(UV) \,, \tag{8}$$

assuming that $\tau$ is sufficiently small. The integrand is now (up to a constant) the density of two independent matrices with iid normal entries. This motivates the following.

**Definition 3.5.** The Normal Product Distribution $NP(\sigma^2)$ is defined as the distribution of the random matrix $X_1 X_2$ where $X_i$ are independent matrices with iid $Normal(\mu = 0, \sigma^2 = \sigma^2)$ entries.

Thus, the Normal Product distribution arises naturally as a "softened" version of the Nuclear Norm distribution. But is there a more precise formal relationship between these distributions? The $1 \times 1$ case already provides a non-trivial and instructive example, while avoiding the notational heaviness of the general case, so we will consider this case in detail before stating the general result.

Letting $X_1$ and $X_2$ be independent centered (scalar) normal random variables, their product $Z = X_1 X_2$ has the following density:

$$P(Z = z) = \frac{1}{2\pi} \int_{(x,y) \in S_z} e^{-x^2/2} e^{-y^2/2} \frac{1}{\sqrt{x^2 + y^2}} dS_z \,. \tag{9}$$

This is well-known Gaunt (2022), and also an easy consequence of the standard Coarea formula (cf. Section B). Here $S_z$ is the hyperbola consisting of all $(x, y)$ whose product is $z$, $dS_z$ is the intrisic length element along this hyperbola, and the $\sqrt{x^2 + y^2}$ factor arises as the Frobenius norm of the Jacobian matrix of the mapping $(x, y) \mapsto xy$.

To evaluate the integral, we can parametrize (one half of) the hyperbola by $x(t) = \sqrt{z}/t, y(t) = \sqrt{z}t$ for $t > 0$. (By symmetry, it is sufficient to assume that $z > 0$, and also to only integrate over one branch of the hyperbola). Then by standard calculus, the length element in this parameterization is given by $dS_z(t) = \sqrt{x'(t)^2 + y'(t)^2} dt = \sqrt{z}\sqrt{1 + t^{-4}}$, so plugging in, we obtain the following one-dimensional integral:

$$P(Z = z) = \frac{1}{\pi} \int_0^\infty e^{-z(t^{-2} + t^2)/2} \sqrt{\frac{1 + t^{-4}}{t^{-2} + t^2}} dt \,; \tag{10}$$

the extra factor of two is from the symmetric integral over the negative branch of the hyperbola.

The formulas at this point are exact, but let us now consider the approximate behavior of the density as $z \to \infty$. Defining $g(t) := (t^{-2} + t^2)/2$, it is easy to see that $g(t)$ has a unique minimum at $t_0 = 1$. We can thus use the standard Laplace approximation to write:

$$P(Z = z) \quad \sim \quad \frac{1}{\pi} \sqrt{\frac{2\pi}{zg''(t_0)}} \sqrt{\frac{1 + t_0^{-4}}{t_0^{-2} + t_0^2}} e^{-zg(t_0)} = \frac{1}{\sqrt{2\pi z}} e^{-z} \,. \tag{11}$$

We can corroborate this result by noting that in this one-dimensional case, the density of $Z$ is exactly known to be $\frac{1}{\pi} K_0(|z|)$, with $K_0$ the modified Bessel function of second kind Johnson & Kotz (1970), and the above analysis recovers the well-known asymptotic $K_0(z) \sim \sqrt{\frac{\pi}{2z}} e^{-z}$ (see e.g. Chapter 5 of Bowman (2010)).

Now how does this compare to the Nuclear Norm distribution? In the $1 \times 1$ case, it is clear that the Nuclear Norm distribution coincides with the classical Laplace distribution with density $\frac{1}{2} e^{-|z|}$. Therefore, the Normal Product distribution does indeed capture the dominant asymptotic factor of $e^{-z}$, however it also has lower order correction terms.

The analysis for the general case is conceptually similar but much more computationally involved. We state the result here for reference and defer the proof to Appendix F.

**Theorem 3.6.** *If $X$ is a $n \times n$ matrix distributed according to the normal product distribution $NP(1)$, with $n > 1$, and $\tau > 0$, then the asymptotic behavior of the density $P(X/\tau)$ as $\tau \to 0$ is given by*

$$P(X/\tau) \sim F(X) \tau^{-3n^2/4 + n/4} e^{-\|X\|_*/\tau} \,, \tag{12}$$

*where $F(X)$ is a certain explicit expression of the singular values of $X$, given in Appendix F.*

Note there is actually a qualitative difference between the $n = 1$ and $n > 1$ cases; in the $n > 1$ case it turns out that the Hessian matrix is not invertible and so it is necessary to restrict the dimensionality to apply the Laplace expansion. Thence the factor of $3/4$ in the exponent of $\tau$.

Contrasting the above result with the Nuclear Norm density, we have $P_{\text{NND}}(X/\tau) = C'\tau^{-n^2} e^{-\|X\|_*/\tau}$. So loosely speaking, the Normal Product distribution looks like the Nuclear Norm distribution with the dimensionality "scaled down" by a factor of $3/4$ (for reasonably large $n$, the $n^2$ term will dominate the exponent). This suggests applying a correction factor of $4/3$ to the

Normal Product, which can be absorbed into the variance, thus arriving at the original approximation we proposed at the beginning of this section. We will see in Section 5.5 that this approximation is quite close for practical purposes.

As mentioned by Srebro et al. (2004); Wipf (2016), the variational characterization of the nuclear norm implies that the MAP estimates when using an NPD prior will coincide with the corresponding estimate under an NND prior, for any likelihood. However, the similarity of the overall distributions has not to our knowledge been formally studied. While the $1 \times 1$ analysis already shows they are not exactly equal, they are indeed closely related as shown by Theorem 3.6.

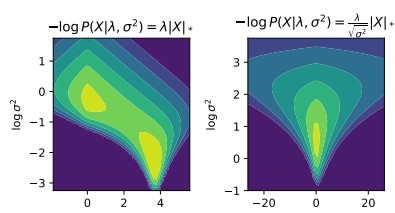

## 4 SIMULATION AND BAYESIAN INFERENCE

We now discuss Bayesian computation under the NND prior.

### 4.1 SAMPLING WITH PROXIMAL LANGEVIN

For inference on as high dimensional an object as a matrix, classical random-walk Metropolis-within-Gibbs style algorithms will be prohibitively slow. However, many modern hybrid Monte Carlo (HMC) methods based on Langevin diffusions Roberts & Tweedie (1996) or Hamiltonian dynamics Duane et al. (1987); Neal (2012) rely on the smoothness of the posterior, structure unavail-

Figure 2: Conditional Prior Ensures Unimodality. x-axis gives first singular value of matrix $X$, y-axis gives estimated error variance, contours indicate posterior density. See Appendix H.2.

able to a posterior based on a Nuclear Norm Distribution prior. Pereyra (2016) proposed a proximal Langevin method to sample from such distributions which will be of use here. This begins by randomly initializing $X$, and then proposing according to the distribution: $P(X^*|X^t) = N(\text{prox}_{\delta/2}^{-\log P}(X^t), \delta\mathbf{I})$. Here, $\text{prox}_{\delta/2}^{-\log P}$ is the proximal operator of the distribution from which samples are desired, $\delta$ is a tunable proposal scale parameter, $X^t$ is the current state of the Markov chain, and $X^*$ is the next proposed state. We set $\delta$ via a Robbins-Monro search Robbins & Monro (1951); Atchadé & Rosenthal (2005) so as to set the acceptance rate to 0.574, which is under certain conditions the optimal rate for proximal MCMC Crucinio et al. (2023). Appendices J.3 and J.4 give the specific implementations for the posteriors used in our numerical applications of Section 5.

In addition to posterior sampling, sampling from the NND prior may also be achieved within this proximal MCMC framework via the proposal $P(X^*|X^t) = N(\text{prox}_{\delta/2}^{\lambda\|\cdot\|_*}(X^t), \delta\mathbf{I})$, where the proximal operator of $\|\cdot\|_*$ is used directly.

### 4.2 POSTERIOR UNDER GAUSSIAN LIKELIHOOD

We now narrow our focus to the case with an isotropic Gaussian likelihood, that is, $Y|X, E = X + E$, where $E_{i,j}|\gamma^2 \sim N(0, \gamma^2)$, $X|\gamma^2 \sim \text{NND}(\frac{\lambda}{\sqrt{\gamma^2}})$, and $\gamma^2 \sim P_{\gamma^2}$. The scaling of $\frac{1}{\sqrt{\gamma^2}}$ is necessary on the prior density in order to achieve a unimodal posterior, similar to the Bayesian Lasso case Park & Casella (2008), as illustrated in Figure 2 and formalized in the following theorem, which is proven in Appendix H.2.

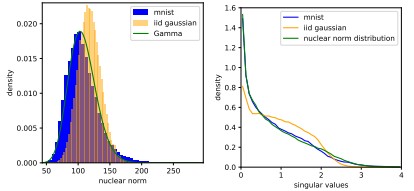

Figure 3: Empirical distributions of matrix nuclear norm (left) and singular values (right) from MNIST dataset (blue) compared to nuclear norm (green) and iid Gaussian (orange).

**Theorem 4.1.** *Let $\lambda > 0$ be fixed and $Y \sim N(X, \gamma^2\mathbf{I})$, and $P_{\gamma^2}$ be as specified in Appendix H.2. Under the conditional prior $P(X|\gamma^2, \lambda) = \text{NND}(\frac{\lambda}{\sqrt{\gamma^2}})$, the posterior $P(X, \gamma^2|Y, \lambda)$ is unimodal. Under the unconditional prior $P(X|\lambda) = \text{NND}(\lambda)$, $P(X, \gamma^2|X, \lambda)$ may be multimodal.*

We now consider how a Gibbs sampler can be constructed for this posterior based on the decomposition given in Theorem 3.4. Reparameterizing from $X$ to $U, \text{diag}(\boldsymbol{\sigma}), V$ leads to the following conditional posteriors:

$$P(\boldsymbol{\sigma}|\cdot) \propto e^{-\lambda \sum_i \sigma_i - \frac{\|\boldsymbol{\sigma}-\mathbf{s}\|_2^2}{2\gamma^2}} \prod_{i \le n} \sigma_i^{m-n} \prod_{i<j\le n} |\sigma_i^2 - \sigma_j^2|.$$

$$P(U|\cdot) \propto e^{-\frac{1}{2\sigma^2}\mathrm{tr}[(YV\mathrm{diag}(\boldsymbol{\sigma}))^\top U]}.$$

$$P(V|\cdot) \propto e^{-\frac{1}{2\sigma^2}\mathrm{tr}[(\mathrm{diag}(\boldsymbol{\sigma})Y^\top V^\top)^\top U]},$$

where $\mathbf{s} = \mathrm{diag}(U^\top YV)$. The posterior conditionals for the unitary matrices take the form of Matrix von-Mises-Fisher distributions Jupp & Mardia (1979). Hoff (2009) proposes an efficient sampler for such distributions which we can exploit as part of a Gibbs sampler iterating through these three conditionals. We perform the $\boldsymbol{\sigma}$ updates via a Metropolis-adjusted Langevin method, which has to deal only with $n$ parameters now rather than $m \times n$ and without any posterior nonsmoothness.

### 4.3 BAYESIAN ESTIMATION OF $\lambda$

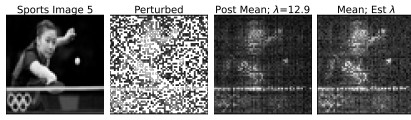

Ignorance of the NND's normalizing constant does not prevent using it as a prior and simulating from its posterior, which was already done by Pereyra (2016). However, it does prevent the analyst from performing Bayesian inference on $\lambda$. In this section, we exploit the closed form expression for the density of the NND of Proposition 3.3 to build an adaptive method which marginalizes over our uncertainty of $\lambda$.

Figure 4: Example matrix completion image from the `sports` dataset; from left to right, the original image, the image with half of pixels removed and noise added, the posterior mean with a fixed $\lambda$ at the optimal value, and the posterior mean of the adaptive method. See Appendices J.4 and J.2 for more.

It's clear from the form that a Gamma prior will be conjugate for $\lambda$, which allows for efficient computation. However, the Gamma prior is not ideal if we want to allow for the possibility of $\lambda$ being at various orders of magnitude Gelman (2006), and we use a half Cauchy instead Polson & Scott (2012) as in the horseshoe prior Carvalho et al. (2009). If we allow for an augmented variable, this can also be implemented using closed form updates using the representation of a Half Cauchy as a hierarchy of inverse Gamma distributions Makalic & Schmidt (2015). That is, if $z_1|z_2 \sim IG(1/2, 1/z_2)$ and $z_2 \sim IG(1/2, 1)$, then $z_1$ is marginally Half Cauchy distributed (ibid). For conjugacy purposes, we will want to sample the inverse of $\lambda$. This leads to the following iteration for sampling from the posterior conditional for $\lambda$ given $X$:

$\beta|\alpha \sim IG(1, 1 + 1/\alpha).$

$\alpha|\beta, X\gamma^2 \sim IG(MN + 1/2, \frac{\|X\|_*}{\sqrt{\gamma^2}} + 1/\beta).$

$\lambda|\alpha \leftarrow \frac{1}{\alpha}.$

Subsequently, $X$ may be simulated conditional on $\lambda$ as discussed either in Section 4.1 or Section 4.2.

## 5 NUMERICAL EXPERIMENTS

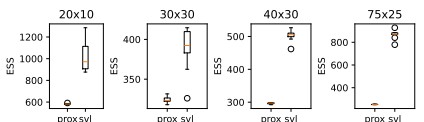

We now conduct numerical experiments illustrating our theory and methodology.

### 5.1 EMPIRICAL BEHAVIOR OF SPATIAL FREQUENCIES OF IMAGES

Figure 5: Effective sample size of proximal (`prox`) versus SVD-Langevin MCMC in (`svl`); higher is better.

Our first simulation is a simple demonstration that the Nuclear Norm distribution can naturally appear "in the wild." We generated a set of matrices by taking the 2-dimensional Fourier transform of MNIST images[1]. Since the high frequencies tend to be near zero, we kept only the $5 \times 5$ submatrix of the Fourier transform corresponding to the lowest frequencies. More details are given in Appendix J.1. This gives a set of $70,000$ images of size $5 \times 5$. We then fit a Nuclear Norm distribution to these

---

[1]Note that we would not expect raw images to be well-fit by the Nuclear Norm distribution, since they tend to have strong correlations between adjacent pixels, whereas the Nuclear Norm Distribution is invariant under any permutation of rows and columns.

| Metric | Dataset Method | butterfly | cityscape | crops | intel | nature | satellite | sports | weather |
|---|---|---|---|---|---|---|---|---|---|
| IntScore | nnd | 0.90 | 0.82 | 0.86 | 0.81 | 0.87 | 0.77 | 0.85 | 0.84 |
| | npd_0.0 | 2.88 | 1.46 | 1.96 | 1.31 | 3.35 | 0.82 | 1.79 | 1.19 |
| | npd_0.33 | 0.76 | 0.69 | 0.73 | 0.69 | 0.72 | 0.65 | 0.73 | 0.70 |
| | npd_0.67 | **0.82** | **0.75** | **0.79** | **0.75** | **0.78** | **0.71** | **0.78** | **0.77** |
| | npd_1.0 | 0.85 | 0.78 | 0.82 | 0.78 | 0.81 | 0.73 | 0.81 | 0.80 |
| | yuchi_0.0 | 5.27 | 3.75 | 4.34 | 3.48 | 5.00 | 3.01 | 3.95 | 3.23 |
| | yuchi_0.03 | 4.18 | 3.16 | 3.97 | 3.41 | 4.15 | 3.40 | 3.53 | 3.26 |
| | yuchi_0.07 | 4.18 | 3.82 | 4.29 | 4.20 | 4.26 | 4.23 | 3.98 | 3.86 |
| | yuchi_0.1 | 4.48 | 4.33 | 4.67 | 4.50 | 4.56 | 4.70 | 4.45 | 4.37 |
| MSE | nnd | 0.18 | 0.16 | 0.17 | 0.16 | 0.17 | 0.15 | 0.17 | 0.17 |
| | npd_0.0 | 0.19 | **0.13** | 0.15 | **0.13** | 0.18 | **0.10** | 0.15 | **0.12** |
| | npd_0.33 | **0.15** | **0.13** | **0.14** | **0.13** | **0.14** | 0.12 | **0.14** | 0.13 |
| | npd_0.67 | 0.16 | 0.14 | 0.15 | 0.14 | 0.15 | 0.13 | 0.15 | 0.14 |
| | npd_1.0 | 0.17 | 0.15 | 0.16 | 0.15 | 0.16 | 0.13 | 0.15 | 0.15 |
| | yuchi_0.0 | 0.19 | 0.14 | 0.15 | **0.13** | 0.17 | 0.11 | 0.15 | **0.12** |
| | yuchi_0.03 | 0.16 | **0.13** | 0.15 | 0.14 | 0.16 | 0.13 | **0.14** | 0.13 |
| | yuchi_0.07 | 0.17 | 0.16 | 0.18 | 0.17 | 0.17 | 0.17 | 0.17 | 0.16 |
| | yuchi_0.1 | 0.19 | 0.18 | 0.19 | 0.19 | 0.19 | 0.19 | 0.19 | 0.19 |
| Time | nnd | 185.76 | 187.84 | 194.63 | 250.86 | 808.70 | 148.52 | 177.68 | 149.26 |
| | npd_0.0 | 3.12 | 2.50 | 3.25 | 3.68 | 5.31 | 1.66 | 2.74 | 1.66 |
| | npd_0.33 | 12.15 | 10.99 | 12.55 | 14.82 | 45.64 | 8.98 | 11.24 | 9.04 |
| | npd_0.67 | 26.03 | 24.54 | 26.73 | 31.91 | 112.56 | 21.38 | 24.76 | 21.51 |
| | npd_1.0 | 43.49 | 42.54 | 44.95 | 54.40 | 209.92 | 38.83 | 42.39 | 39.07 |
| | yuchi_0.0 | 93.77 | 107.80 | 151.04 | 129.74 | 127.57 | 81.35 | 82.01 | 85.42 |
| | yuchi_0.03 | 137.73 | 98.48 | 114.83 | 82.10 | 179.25 | 105.09 | 89.57 | 93.32 |
| | yuchi_0.07 | 139.17 | 119.65 | 147.06 | 136.66 | 301.37 | 143.39 | 122.62 | 109.61 |
| | yuchi_0.1 | 133.29 | 127.48 | 114.23 | 114.22 | 377.63 | 109.52 | 138.78 | 99.14 |

Table 1: **Comparative Results Across Different Bayesian Models.** Each column gives a dataset, each set of rows gives a metric, and individual rows give a particular method. `nnd` refers to the adaptive nuclear method while `npd` refers to the normal product distribution. We include Yuchi et al. (2023) as a baseline. The decimal after either `npd` or `yuchi` refers to the rank of the approximation, given as a proportion of the maximum possible rank. `IntScore` refers to the Interval Score of Gneiting & Raftery (2007), which scores a prediction interval both on its width and on its distance to the true value. These are generated from the Bayesian model using a equi-probable tailed credible interval based on the MCMC chain. We observe that the nuclear norm based models in general are able to produce much more accurate intervals. Furthermore the `npd` approach using about two thirds of the rank consistently does best. However, when it comes to MSE of the posterior mean, the differences are much smaller, with different methods achieving the best score on different problems. The Normal Product distribution is consistently the fastest to execute, and though these figures seem to suggest that the approach of Yuchi et al. (2023) is more computationally efficient than the proximal MCMC used for the `nnd` approach, note that the `yuchi` method is constrained to only using at most a tenth of the total possible matrix rank due to memory constraints, and would seemingly require more time for larger ranks.

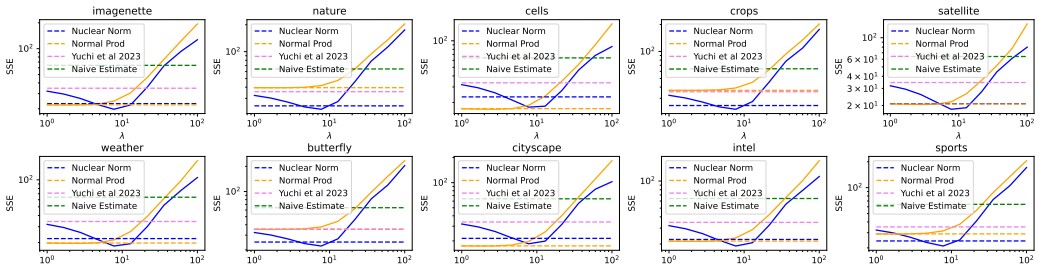

Figure 6: SSE of Bayesian-estimated $\lambda$ (horizontal green line) vs a range of fixed $\lambda$'s (blue line) on Matrix Denoising problems; dotted orange line gives naive estimate; lower is better.

matrices. Here there is only one free parameter, namely the scale factor $\lambda$; fitting this amounts to matching the data's average nuclear norm.

In Figure 3 we compare the empirical MNIST distribution with the fitted Nuclear Norm distribution. We see a remarkably close fit in terms of both the Nuclear Norm density as well as the density of individual singular values. For comparison, we also fit an alternative random matrix model in which the entries are iid centered Gaussians. As also shown in the figure, the fit of this model to the MNIST frequencies is considerably worse than the fit of the Nuclear Norm distribution.

## 5.2 Effective Sample Size for Normal Sampler

To study the potential for the sampler of Section 4.3 to provide improved mixing of the Markov chain under Gaussian likelihoods, we conduct a simulation study. We generate synthetic rank 1 matrices from the normal product distribution. Subsequently, we run a Proximal Langevin sampler and a sampler based on the SVD. For both samplers, we automatically adjust the step size, aiming for an acceptance rate of $0.574$ with $10,000$ sampling iterations and a burn in of $1,000$. Figure 5 gives the results. On the left side of the figure is the Effective Sample Size (ESS), a measure of chain quality which discounts correlations to estimate the size of an independent sample that a correlated sample is equivalent to (Geyer, 2011). We see that the sampler based on Theorem 3.4 gives consistently better ESS than the proximal Langevin sampler of (Pereyra, 2016). The performance gap is highest on the $75 \times 25$ sized problem.

## 5.3 Matrix Denoising

We now test the Bayesian $\lambda$ estimation of Section 4.3/Appendix J.3 on ten datasets of natural images (see Appendix J.2). We ran the sampler with $\lambda$ fixed at 10 values along a logarithmic grid from $0.01$ to $100$, as well as using the adaptive Horseshoe method. We measure performance in terms of Mean Squared Reconstruction Error. The results are given in Figure 6. The solid blue line gives the performance of the method along the grid of $\lambda$ values, while the green dotted horizontal line gives the performance of the adaptive method. The dotted orange line gives the performance of the Naive method which simply predicts using the observed noisy pixels. We see that the adaptive method is able to achieve prediction error in line with the optimal fixed $\lambda$ value across all datasets.

## 5.4 Noisy Matrix Completion

We use the same image datasets for the matrix completion benchmark. We randomly and independently delete pixels (meaning that their value is not observable to the sampler) each with probability $0.5$, and again add iid Gaussian noise with a standard deviation of $0.1$ to the observed entries. We extend the low-rank denoising algorithm of Pereyra (2016) to accommodate this matrix completion problem (see Appendix J.4), and again compare the $\lambda$-estimation procedure to fixed a grid of fixed $\lambda$. The mean square completion error is given in Figure 7. Again the adaptive method consistently performs as well as the optimal value from the grid.

## 5.5 Comparison with Normal Product Spectrum

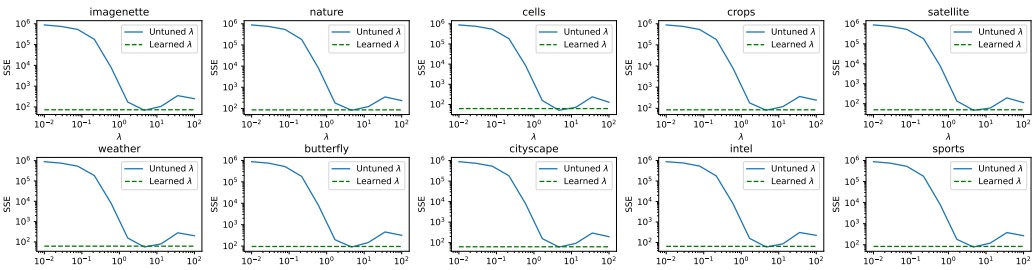

Figure 7: SSE of Bayesian-estimated $\lambda$ (horizontal green line) vs a range of fixed $\lambda$'s (blue line) on Matrix Completion problems; lower is better.

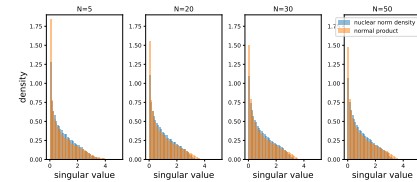

Motivated by the results in Section 3.4, we here explore the quality of the approximation $\mathrm{NND}(\lambda = 1) \approx NP(\sigma^2 = 4/3)$. Note that we also provide more details on known analytic results for the spectrum of $NP$ in Appendix G.

We generated samples from the Nuclear Norm distribution using the proximal Langevin method of Section 4.1. For both the Nuclear Norm samples, and the Normal

Figure 8: Empirical singular value density, for $\mathrm{NND}(1)$ and $NP(\sigma^2 = 4/3)$ samples.

product samples, we rescaled by $1/n$ (the size of the matrix) in order to ensure a consistent limit (for these simulations, we only consider square matrices). For each matrix size, and each of the two distributions we generated 20000 samples.

In Figure 8 we show the empirical distribution of singular values of the two distributions. The Normal Product places somewhat more mass on the smallest singular values, although the distributions otherwise track each other quite closely. Further comparisons are shown in Appendix I. Overall, the Normal Product distribution provides a reasonably good first approximation to the Nuclear Norm Distribution, but there are some notable "second order" statistical deviations that do not appear to abate with increasing dimension.

## 6 Discussion

The nuclear norm distribution has somehow escaped an analysis of its basic properties, which we established in this article. This included its normalizing constant, the distribution of the nuclear norm itself, the distribution of its singular values, and its close relationship to the (considerably simpler and more tractable) normal product distribution. Subsequently, we demonstrated how this knowledge could be deployed to improve some Bayesian low-rank inference algorithms. As a means of conducting matrix denoising and completion, the results of Section 5.3 revealed that Bayesian estimation of $\lambda$ was able to consistently achieve error on par with the optimal penalty term from a grid. In many real world applications, it would not be possible to perform this grid search due to a lack of ground-truth data, making the adaptive method especially crucial.

By specifying the density of the nuclear norm distribution, we have unlocked the ability to deploy Bayesian hierarchical modeling on the penalty parameter $\lambda$. We exploited this ability by automatically inferring $\lambda$, but there are many more possibilities for complex data. For example, matrices observed over time and space, such as that from satellite data, could be modeled as coming from a spatiotemporal random process. Additionally, there are related distributions beyond the scope of this article which demand future study. For instance, practitioners in low rank optimization have also developed "weighted nuclear norm" Mohan & Fazel (2012) methods which use a modification of the nuclear norm with a different penalty weight for each singular value.

### Reproducibility Statement

This article is accompanied with a zip file containing Python code which can be used to reproduce the figures herein. All proofs are given below in the supplementary material.

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

# A    PROXIMAL OPERATORS FOR NONSMOOTH OPTIMIZATION AND SIMULATION

Proximal operators are objects from convex analysis (Rockafellar, 1997) which are useful to analyze composite functions Parikh et al. (2014) in machine learning Polson et al. (2015) which are given as the sum of two functions:

$$\min_{x \in \mathcal{X}} c(x) = f(x) + g(x), \tag{13}$$

where $f$ is a complicated differentiable function and $g$ is a simple but nonsmooth function. In particular, we will be interested in the proximal operator associated with $g$, which maps domain of $g$, in this case $\mathcal{X}$, to itself. Given some norm on $\mathcal{X}$, the action of the proximal operator on some vector $x$ is defined as:

$$\mathrm{prox}^g(x) = \underset{u \in \mathcal{X}}{\mathrm{argmin}} \; \frac{\|u - x\|^2}{2} + g(u). \tag{14}$$

At an intuitive level, given an input $x$, a proximal operator finds us some other point in the domain which is not too far away from $x$ but does a better job at minimizing $g$. In this article, we will be using standard Euclidean norms divided by some positive constant $s$. The proximal operator will then be denoted as:

$$\mathrm{prox}_s^g(x) = \underset{u \in \mathcal{X}}{\mathrm{argmin}} \; \frac{\|u - x\|^2}{2s} + g(u), \tag{15}$$

where $\|\cdot\|$ is either the vector 2 norm or matrix $F$ norm for the purposes of this article.

Proximal operators are most interesting in practice when there is a straightforward way to compute their action on an arbitrary vector, either in closed form or via some efficient iteration. Perhaps the most well known proximal operator is that of the $\ell_1$ norm, which is given by:

$$\mathrm{prox}_s^{\lambda \|\cdot\|_1}(x) = \underset{u \in \mathbb{R}^M}{\mathrm{argmin}} \; \frac{\|u - x\|_2^2}{2s} + \lambda\|u\|_1. = \mathrm{sgn}(x)(x - s\lambda)^+, \tag{16}$$

where sgn is the sign function and $(\cdot)^+$ is the positive part function, which returns its argument if it is positive and zero otherwise, each applied elementwise on the vector $x$. This is called the *Soft Thresholding Operator*, or STO. Similarly, the proximal operator associated with the nuclear norm has an action computable in closed form. Namely, it is given by applying the STO to the singular values of the matrix. If written in the language of matrix functions Higham (2008), which extend the definition of scalar functions to define matrix-to-matrix functions by way of applying them individually to singular values of a matrix, this fact may be written using almost exactly the same symbols:

$$\mathrm{prox}_s^{\lambda \|\cdot\|_*}(X) = \underset{U \in \mathbb{R}^{M \times N}}{\mathrm{argmin}} \; \frac{\|U - X\|_F^2}{2s} + \lambda\|U\|_* = \mathrm{sgn}(X)(X - s\lambda)^+. \tag{17}$$

We return now to our analysis of the composite function $c$. Given some point $x$, a proximal operator can be used to define a quantity called a *gradient-like step*, which can often be used in iterative algorithms on nonsmooth functions in places where gradients would be on smooth ones. It is defined as:

$$G^s(x) = \frac{\mathrm{prox}_s^g(x - s\nabla f(x)) - x}{s} \tag{18}$$

For example, plugging this into the gradient descent formula $x^{t+1} \leftarrow x^t - sG^s(x)$ yields the iteration $x^{t+1} \leftarrow \mathrm{prox}_s^g(x - s\nabla f(x))$, which is called proximal gradient descent.

In a seminal article, Pereyra (2016) showed how the proximal operator of the negated log density could be used to accelerate sampling from nonsmoooth posterior distributions. In particular, they recommend to use as a proposal density for a Metropolis-Hastings algorithm the following distribution:

$$P(X^*|X^t) = N(\mathrm{prox}_{\delta/2}^{-\log P_{X|Y}}(X^t), \delta \mathbf{I}) \tag{19}$$

where $\delta$ determines the scale of the diffusion and $-\log P_{X|Y}$ is the log posterior density.

## B  Coarea Formula and Pushforward of Measure

We will make extensive use of the Smooth Coarea Formula, which does not appear to be well known in the machine learning community in proportion to its usefulness. General references for the material in this section are Chavel (2006) and Segert & Davis-Stober (2019).

In brief, this formula is a simultaneous generalization of the standard Change of Variables Formula and Fubini's Theorem. Whereas the former tells us how to modify an integral under an invertible, non-linear coordinate mapping, and the latter tells us how to modify it under a non-invertible, linear mapping, the Coarea Formula handles the general case of a smooth *non-invertible, non-linear*, mapping.

Suppose we have a smooth mapping $F : \mathbb{R}^m \to \mathbb{R}^n$, with $m \geq n$. Denote its Jacobian matrix at the point $x \in \mathbb{R}^m$ by $J_x$; this is a matrix of size $m \times n$. We require that $F$ is a *submersion*, which means that the Jacobian matrix at each point is full rank. Equivalently, $J_x^T J_x$ is invertible for each $x$.

Then the following holds for any integrable $f$:

$$\int_{x \in \mathbb{R}^m} f(x) \sqrt{\det(J_x^T J_x)} dx = \int_{y \in \mathbb{R}^n} \left( \int_{z \in F^{-1}y} f(z) V_{F^{-1}y}(dz) \right) dy. \tag{20}$$

Here $F^{-1}$ gives the preimage, i.e. $F^{-1}y = \{z \in \mathbb{R}^m : F(z) = y\}$, and $V_{F^{-1}y}$ is the intrinsic volume measure on $F^{-1}y$, which is the Riemannian restriction of the standard Euclidean volume form. The set $F^{-1}y$ is guaranteed to be a smooth manifold of dimesnion $m - n$ (and thus to have a well-defined volume form) by the classical Implicit Function Theorem. If we have a set of local coordinates $u : U \mapsto F^{-1}y$, which recall is a smooth invertible map from an open set $U$ in Euclidean space to an open set of $F^{-1}y$, then the volume form takes the local form

$$V_{F^{-1}y}(dz(u)) = \sqrt{\det_{ij} \langle u_i', u_j' \rangle} du, \tag{21}$$

where $du$ is now the Euclidean volume form on the open set $U$.

The Coarea Formula also holds when $\mathbb{R}^m$ and $\mathbb{R}^n$ are replaced with general Riemannian manifolds, after making the necessary modifications.

Next, we remark on the connection between the Coarea Formula and the general problem of characterizing the pushforward of a measure.

In general, given a measure $\mu$ on some measure set $A$ and a measurable mapping $G : A \to B$, the pushforward $G_*\mu$ is the measure on $B$ defined as follows, for any measurable set $S$ on $B$:

$$(G_*\mu)(S) = \mu(G^{-1}S). \tag{22}$$

If $\mu$ is the distribution of a random variable $X$, then $G_*\mu$ is the distribution of the random variable $G(X)$.

In particular, we have the change of variables formula:

$$\int_B f(y) d(G_*\mu)(y) = \int_A f(F(x)) d\mu(x) \tag{23}$$

for any measurable $f : B \to \mathbb{R}$. Indeed, if $f$ is an indicator function of a measurable set, then this is just a restatement of the definition of the pushforward; by approximating an arbitrary measurable function with a sum of indicators in a standard way, the general formula follows.

Now assume that $A = \mathbb{R}^m$ and $B = \mathbb{R}^n$, that $G$ is a smooth submersion, and furthermore that the measure $\mu$ has a density $d\mu/dx$. What is the density of the pushforward?

Well, given any measurable $g : \mathbb{R}^n \to \mathbb{R}$, we apply the Coarea Formula to the function $\mathbb{R}^m \to \mathbb{R}$, $x \mapsto \frac{d\mu}{dx}(x)g(G(x))\det(J_x^T J_x)^{-1/2}$, obtaining

$$\int_{x \in \mathbb{R}^m} g(G(x)) \frac{d\mu}{dx}(x) dx = \int_{y \in \mathbb{R}^n} \left( \int_{z \in G^{-1}y} g(G(z)) \frac{d\mu}{dx}(z) \det(J_z^T J_z)^{-1/2} V_{G^{-1}y}(dz) \right) dy \tag{24}$$

$$= \int_{y \in \mathbb{R}^n} g(y) \left( \int_{z \in G^{-1}y} \frac{d\mu}{dx}(z) \det(J_z^T J_z)^{-1/2} V_{G^{-1}y}(dz) \right) dy \tag{25}$$

On the other hand, by the change of variables formula,

$$\int_{x \in \mathbb{R}^m} g(G(x)) \frac{d\mu}{dx}(x) dx = \int_{y \in \mathbb{R}^n} g(y) d(G_*\mu)(y) \,. \tag{26}$$

Comparing the two, and recalling that $g$ was arbitrary we conclude that $G_*\mu$ has the following density:

$$\frac{d(G_*\mu)}{dy}(y) = \int_{z \in G^{-1}y} \frac{d\mu}{dx}(z) \det(J_z^T J_z)^{-1/2} V_{G^{-1}y}(dz) \,. \tag{27}$$

## C  MATRIX CALCULUS REFRESHER

We will make use below of matrix calculus techniques. Since this has a reputation for being opaque we provide a brief refresher.

Suppose we have a matrix $M(\theta)$ that depends smoothly on some vector of parameters $\theta \in \mathbb{R}^p$. Denote the tangent vectors with respect to each component by $\partial\theta_i$. For our purpose, it is sufficient to regard these as elements of a formal vector space.

The differential $dM$ is defined as the following (linear) mapping of the tangent vectors:

$$dM(\partial\theta_i) := \frac{\partial M}{\partial\theta_i} \,, \tag{28}$$

or, even more explicitly,

$$(dM(\partial\theta_i))_{ab} := \frac{\partial M_{ab}}{\partial\theta_i} \,. \tag{29}$$

The main usefulness is that $d$ satisfies the Matrix Leibniz rule:

$$d(MN) = (dM)N + M(dN) \,. \tag{30}$$

But what does this actually mean, given that $dM$ and $dN$ are not matrices? It really means that if we evaluate both sides at any tangent vector, then they coincide:

$$d(MN)(\partial\theta_i) = dM(\partial\theta_i)N + MdN(\partial\theta_i), \forall i \in \{1, \ldots, p\} \,. \tag{31}$$

In this equation, $dM(\partial\theta_i)$ and $dN(\partial\theta_i)$ are bonafide matrices so there is no difficulty in interpreting the products.

In practice, we would have some function:

$$F(M) = \text{some expression involving various matrix operations.}$$

Then by applying properties of $d$, we would get:

$$dF(M) = \text{some other expression involving various matrix operations and } dM.$$

Now if we want to actually back out explicit expressions for the partial derivatives $\partial F(M(\theta))/\partial\theta_i$, we would evaluate both sides of the above on the tangent vector $\partial\theta_i$. Operationally, this amounts to replacing every occurrence of $dM$ on the right hand side with the matrix $\frac{\partial M}{\partial\theta_i}$, resulting in some matrix expression which is equal to $\partial F(M(\theta))/\partial\theta_i$.

Finally, one common situation is when each entry of $M$ is an independent variable. In this case, it is easy to see that the derivative of $M$ with respect to the $i, j$ component is the one-hot matrix $e_{ij}$. This fact will be used later without comment.

## D  NORMALIZING CONSTANT PROOF

We prove here the formula for the normalizing constant $C(\lambda) = \int_{\mathbb{R}^{n \times m}} e^{-\lambda \|M\|_*} dM$. By homogeneity of the norm, it is no loss of generality to take $\lambda = 1$. Further, it is no loss of generality to assume $n \leq m$.

In this case, we can write the singular value decomposition of a given matrix $M$ as $M = U\text{diag}(\sigma)V^T$, where $U \in \mathbb{R}^{n \times n}$, $\sigma \in \mathbb{R}^n_{\geq 0}$ and $V \in \mathbb{R}^{m \times n}$. Moreover $U$ is orthogonal and $V$ has orthogonal columns; that is $V^T V = I_n$.

It is easy to see that $U, V, \sigma$ can be chosen to vary smoothly as a function of sufficiently small perturbations of $M^2$. Thus we can differentiate both sides of the equation $M = U\text{diag}(\sigma)V^T$ with respect to the entries of $M$. By also differentiating the orthogonality constraints we obtain the following system of constraints:

$$
\begin{align}
dM &= dU\text{diag}(\sigma)V^T + U\text{diag}(d\sigma)V^T + U\text{diag}(\sigma)dV^T\,, \tag{32}\\
0 &= dU^T U + U^T dU\,, \tag{33}\\
0 &= dV^T V + V^T dV\,. \tag{34}\\
& \tag{35}
\end{align}
$$

Multiply the top equation by $U^T$ on the left and $V$ on the right:

$$
\begin{align}
U^T dMV &= U^T dU\text{diag}(\sigma)V^T V + U^T U\text{diag}(d\sigma)V^T V + U^T U\text{diag}(\sigma)dV^T V \tag{36}\\
&= U^T dU\text{diag}(\sigma) + \text{diag}(d\sigma) + \text{diag}(\sigma)dV^T V\,. \tag{37}
\end{align}
$$

By the constraint equations, we know that $U^T dU$ and $dV^T V$ are both anti-symmetric, so in particular will have all zero entries along the diagonal. Thus the product of one of these matrices with a diagonal matrix also has all zeros along the diagonal (in particular the trace vanishes). So taking the trace of both sides of the above, we get $\text{tr}(U^T dMV) = \text{tr}(\text{diag}(d\sigma))$. On the other hand, evidently $\text{tr}(\text{diag}(d\sigma)) = d\|M\|_*$, so evaluating both sides on the tangent vector $\partial M_{ij}$ we get the following formula:

$$
\begin{align}
\frac{\partial \|M\|_*}{\partial M_{ij}} &= \text{tr}(U^T e_{ij} V) \tag{38}\\
&= \text{tr}(VU^T e_{ij}) \tag{39}\\
&= (UV^T)_{ij}\,, \tag{40}
\end{align}
$$

or, in matrix notation,

$$
\frac{\partial \|M\|_*}{\partial M} = UV^T\,. \tag{41}
$$

Returning to the normalizing constant, we will apply the Coarea formula to the mapping $M \mapsto \|M\|_*$. The Jacobian $J_x$ is in this case a $(mn) \times 1$ matrix so the determinant becomes

$$
\begin{align}
\det(J_M^T J_M) &= \sum_{ij} \left(\frac{\partial \|M\|_*}{\partial M_{ij}}\right)^2 \tag{42}\\
&= Tr\left(\left(\frac{\partial \|M\|_*}{\partial M}\right)^T \frac{\partial \|M\|_*}{\partial M}\right) \tag{43}\\
&= \text{tr}(VU^T UV^T) \tag{44}\\
&= \text{tr}(V^T V) \tag{45}\\
&= n\,. \tag{46}
\end{align}
$$

Applying the Coarea Formula yields

$$
\begin{align}
\sqrt{n} \int e^{-\|M\|_*} dM &= \int_{\mathbb{R}_{\geq 0}} \left(\int_{M' \in S^*(t)} e^{-\|M'\|_*} V_{B^*(t)}(dM')\right) dt \tag{47}\\
&= \int_{\mathbb{R}_{\geq 0}} e^{-t} \left(\int_{M' \in S^*(t)} V_{S^*(t)}(dM')\right) dt\,, \tag{48}
\end{align}
$$

where $S^*(t) = \{M : \|M\|_* = t\}$ is the "unit sphere" with respect to the nuclear norm. The inner integral is simply the $nm-1$-dimensional volume of this sphere (i.e. the "surface area" when regarded

---

$^2$Strictly speaking, this is only true provided that $M$ has distinct singular values; since the set of $M$ that have repeated singular values has measure zero it is no harm to assume we are in the former case

as an $nm - 1$-dimensional submanifold of $\mathbb{R}^{nm}$). By homogeneity of the norm, $S^*(t) = tS^*(1)$. Since the sphere has dimension $nm - 1$ this implies $\text{Vol}_{nm-1}S^*(t) = t^{nm-1}\text{Vol}_{nm-1}(S^*(1))$. So the integral becomes

$$\text{Vol}_{nm-1}(S^*(1)) \int_{\mathbb{R}_{\geq 0}} e^{-t}t^{nm-1}dt = \text{Vol}_{nm-1}(S^*(1))\Gamma(nm) = \text{Vol}_{nm-1}(S^*(1))(nm-1)!, \quad (49)$$

as claimed.

## E  STOCHASTIC FACTORIZATION PROOF

We'll actually prove the more general result given in Anderson (2010), namely that

$$\int \phi(\sigma(M))dM = C_{nm} \int_{\mathbb{R}_+^n} \phi(x)|\Delta(x^2)| \prod_{i=1}^n x_i^{n-m}dx \quad (50)$$

where $\Delta$ is the standard Vandermonde determinant, $\phi$ is any "nice" function, $\sigma(M)$ is the vector of singular values (ordered from largest-to-smallest, say), and $C_{nm}$ is an explicit constant only depending on $n, m$. For our purposes, the value of the constant is not so important, so we won't explicitly keep track of it, but it can be backed out from the following argument with a bit more work. Accordingly, we'll use the symbol $\sim$ to mean two quantities are equal up to a multiplicative constant which only depends on $n, m$.

While this result is proved in Anderson (2010), their proof is quite technical and involved, while the following uses only multivariate calculus. We think the following proof is probably well-known, but we don't know a reference so we record it here.

Equation 50 will follow immediately from the standard multivariate change-of-variables formula, provided that we can show

$$\det\left(\frac{d\text{Prod}}{d(U, s, V)}\right) \propto \pm\Delta(s^2) \prod_{i=1}^n s_i^{n-m} \quad (51)$$

where Prod denotes the product mapping:

$$\text{Prod} : O(n) \times \mathbb{R}_{+,\leq}^n \times S(n, m) \quad \rightarrow \quad \mathbb{R}^{n \times m} \quad (52)$$
$$\text{Prod}(U, s, V) \quad = \quad U\text{diag}(s)V^T \quad (53)$$

where $R_{+,\leq}$ denotes the set of sequences which are non-negative and non-decreasing.

So it suffices to establish Equation 51. Our strategy for this will be to consider a particular random matrix $W$ for which the density of $W$, as well as the joint density of its singular values/vectors are both known in explicit form. The desired Jacobian determinant is then given as the ratio of the two densities, as per the standard change-of-variables formula.

To wit, define $W$ as a random $n \times m$ matrix with independent unit Gaussian entries. On the one hand, the density of this matrix is evidently given by

$$p_W(X) \sim e^{-\|X\|_F^2/2} \quad (54)$$

(here $X$ is a dummy variable). Consider now the random vector $\lambda_W$ defined as the eigenvalues of $WW^T$ (ordered from largest to smallest, say). Since $WW^T$ has a Wishart distribution with $m$ degrees-of-freedom, the distribution of $\lambda_W$ follows the well-known Wishart spectral density:

$$p_{\lambda_W}(\lambda) \sim \Delta(\lambda) \prod_i e^{-\lambda_i/2}\lambda_i^{(m-n-1)/2}$$

If we define the random vector $s_W$ as the singular values of $W$, ordered as above, then $s_W$ is given as the element-wise square root of $\lambda_W$. Thus applying the standard change-of-variables formula, we obtain the following density:

$$p_{s_W}(s) \sim \Delta(s^2) \prod_i e^{-s_i^2/2}s_i^{m-n}$$

Letting $U_W$ and $V_W$ denote the singular vectors of $W$, it is clear by the rotational symmetry of $W$ that both of these matrices are uniformly distributed over their domains of definition (respectively $O(n)$ and $S(n, m)$).

Conversely, it follows from the above discussion that if we consider random matrices $U \in O(n), V \in S(n, m)$ and a random vector $s \in \mathbb{R}^n_{+,\leq}$ with joint density $p(U, s, V) \sim p_{S_w}(s)$, then the random matrix $\mathrm{Prod}(U, s, V)$ has the same distribution as $W$

Yet again by the standard change-of-variables formula, we conclude that the Jacobian of the mapping is related to the two densities as

$$\det\left(\frac{d\mathrm{Prod}}{d(U, s, V)}\right) = \pm\frac{p_{S_w}(s)}{p_W(\mathrm{Prod}(U, s, V))} \tag{55}$$

which gives the result after plugging in and simplifying.

*Remark* E.1. One might wonder whether this argument is circular, since it relies on the form of the Wishart spectral density, which (one might think) possibly requires knowing the form of the Jacobian to derive in the first place. However it is possible to derive the form of the Wishart spectrum in a way that avoids use of this determinant or something equivalent (cf. https://galton.uchicago.edu/ lalley/Courses/386/ClassicalEnsembles.pdf), so the argument is actualy not circular.

## F  Normal Product Density Asymptotic

Consider matrices $U, V$ with iid unit normal entries, of shape $n \times n$ and set $X = UV$. Let $P(X)$ be the density function. And let $\tau > 0$ be a positive number.

By the Coarea Formula, we have

$$P(X) = \frac{1}{(2\pi\tau)^{n^2}} \int_{S_X} J(U, V)^{-1} e^{(-\|U\|_F^2/(2\tau) - \|V\|_F^2/(2\tau))} dS_X(U, V),$$

where $J(U, V)$ is defined in terms of the Jacobian matrix $Jac$ of the mapping $(U, V) \to UV$:

$$J(U, V) := \det((Jac)(Jac)^T)^{1/2},$$

$S_X := \{(U, V) : UV = X\}$, and $dS_X$ is the intrinsic volume form on $S_X$.

The basic idea is we want to let $\tau \to 0$ and perform a Laplace expansion of the integral. The issue, however, is that the Hessian matrix of the mapping $U, V \mapsto \|U\|_F^2 + \|V\|_F^2$ is not invertible, even when restricted to $S_X$. Already this can be seen in the $2 \times 2$ case by a simple calculation.

Intuitively, this makes sense, because we know the minimizers of the functional must take the form $U_{svd}D^{1/2}, D^{1/2}V_{svd}$, where $U_{svd}$ and $V_{svd}$ are *orthogonal* matrices (namely, the singular vectors of $X$) and $D$ is diagonal.

So define the submanifold

$$S'_X = \{(U, V) \in S_X : U^T U, VV^T \text{ are diagonal}\}. \tag{56}$$

By the above discussion, any minimizers of the functional must lie in $S'_X$. What is less obvious, but will be shown later, is that if $U, V$ are a minimizer, then the tangent space to $S'_X$ at $(U, V)$ is the largest subspace on which the Hessian at $U, V$ is invertible.

Thus, we can still perform the Laplace expansion, but when computing the Hessian, we have to just remember to throw out the "null" dimensions (i.e. the ones orthogonal to the tangent space of $S'_X$), and treat the ambient dimension as $\dim(S'_X)$ rather than $\dim(S_X)$.

When we perform the expansion, we will thus get an expression like the following:

$$P(X) \sim n! 2^n \frac{1}{(2\pi\tau)^{n^2}} (2\pi\tau)^{n(n+1)/4} |Hess(U_{min}, V_{min})|^{-1/2} VF(U_{min}, V_{min}) J(U_{min}, V_{min})^{-1} e^{-\|X\|_*/\tau},$$

$$\tag{57}$$

where $U_{min}$ and $V_{min}$ are any choice of minimizers that attain the nuclear norm. The factor of $n! 2^n$ arises from the other symmetric minimizers obtained by reordering the singular values and/or

multiplying some pairs of singular vectors by $-1$. And $VF$ denote the intrinsic Riemannian volume form on $S_X$ expressed in local coordinates. Note finally the crucial $\tau^{n(n+1)/4}$ factor, with the exponent reflecting the fact that $\dim(S'_X)/2 = n(n+1)/4$.

Finally, by rotational symmetry, it is no loss of generality to assume that $X$ is a diagonal matrix with positive entries, in which case we may take $U_{min} = V_{min} = X^{1/2}$. Let $\{d_i\}_i$ be the diagonal elements (i.e. singular values) of $X$. We'll also assume that all singular values are distinct and strictly positive, which is a full-measure condition and thus harmless for our purpose. We now analyze each of the terms above in turn.

## F.1 HESSIAN COMPUTATION

Let's first consider the Hessian term. The constraint set $S_X$ is defined by $UV = X$, so we can locally parameterize it $W \mapsto (W, W^{-1}X)$, $W \in \mathbb{R}^{n \times n}$ whence the function inside the exponential becomes $F(W) = \|W\|_F^2/2 + \|W^{-1}X\|_F^2/2$, in local coordinates. We can easily compute the first derivatives:

$$
\begin{align}
dF(W) &= d\mathrm{tr}(W^TW)/2 + d\mathrm{tr}(X^T(W^{-1})^TW^{-1}X)/2 \tag{58}\\
&= d\mathrm{tr}(W^TW)/2 + d\mathrm{tr}((WW^T)^{-1}XX^T)/2 \tag{59}\\
&= \mathrm{tr}(W^TdW) + \mathrm{tr}(d((WW^T)^{-1})XX^T)/2 \,. \tag{60}
\end{align}
$$

Breaking out the internal term:

$$
\begin{align}
d((WW^T)^{-1}) &= -(WW^T)^{-1}d(WW^T)(WW^T)^{-1} \tag{61}\\
&= -(WW^T)^{-1}(dWW^T + WdW^T)(WW^T)^{-1} \tag{62}
\end{align}
$$

and plugging back in yields

$$
\begin{align}
dF(W) &= \mathrm{tr}(W^TdW) - \frac{1}{2}\mathrm{tr}((WW^T)^{-1}dWW^T(WW^T)^{-1}XX^T) \tag{63}\\
&\quad - \frac{1}{2}\mathrm{tr}((WW^T)^{-1}WdW^T(WW^T)^{-1}XX^T) \,, \tag{64}\\
dF(W) &= \mathrm{tr}(W^TdW) - \frac{1}{2}\mathrm{tr}(W^T(WW^T)^{-1}XX^T(WW^T)^{-1}dW) \tag{65}\\
&\quad - \frac{1}{2}\mathrm{tr}(((WW^T)^{-1}XX^T(WW^T)^{-1}W)^TdW) \,, \tag{66}\\
(\frac{dF(W)}{dW})^T &= W^T(I - (WW^T)^{-1}XX^T(WW^T)^{-1}) \,. \tag{67}
\end{align}
$$

To get the Hessian, we need to take the differential again of the right hand side:

$$
\begin{align}
H &= dW^T(I - (WW^T)^{-1}XX^T(WW^T)^{-1}) \tag{68}\\
&\quad + W^T(-d((WW^T)^{-1})XX^T(WW^T)^{-1} - (WW^T)^{-1}XX^Td((WW^T)^{-1})) \tag{69}\\
&= dW^T(I - (WW^T)^{-1}XX^T(WW^T)^{-1}) \tag{70}\\
&\quad + W^T(WW^T)^{-1}(dWW^T + WdW^T)(WW^T)^{-1}XX^T(WW^T)^{-1} \tag{71}\\
&\quad + W^T(WW^T)^{-1}XX^T(WW^T)^{-1}(dWW^T + WdW^T)(WW^T)^{-1} \,, \tag{72}
\end{align}
$$

where the indices are such that $H(\partial_{ij})_{i'j'} = \partial^2 W'/\partial_{ij}\partial_{j'i'}$.

Now we can finally evaluate this at $W_{min} = X^{1/2}$ (recall we rotated coordinates so that $X$ is a positive, diagonal matrix). In this case, we evidently have $X = X^T = WW^T$, so the above formula simplifies:

$$
H = X^{-1/2}(dW(X^{1/2})^T + X^{1/2}dW^T) + X^{1/2}(dW(X^{1/2})^T + X^{1/2}dW^T)X^{-1} \,. \tag{73}
$$

Evaluating the tensor (and using symmetry of $X$), we obtain:

$$
\begin{align}
H(\partial_{ij})_{i',j'} &= (X^{-1/2}e_{ij}X^{1/2})_{i'j'} + (e_{ji})_{i'j'} + (X^{1/2}e_{ij}X^{-1/2})_{i'j'} + (Xe_{ji}X^{-1})_{i'j'} \tag{74}\\
&= \delta_{ij=i'j'}(d_i^{-1/2}d_j^{1/2} + d_i^{1/2}d_j^{-1/2}) + \delta_{ij=j'i'}(1 + d_j/d_i) \,. \tag{75}\\
&= (\delta_{ij=i'j'} + d_j^{1/2}d_i^{-1/2}\delta_{ji=i'j'})(d_i^{-1/2}d_j^{1/2} + d_i^{1/2}d_j^{-1/2}) \tag{76}
\end{align}
$$

where we used the fact that, for diagonal matrices, $(D_1 e_{ij} D_2)_{i'j'} = (D_1)_{i'i}(D_2)_{jj'} = \delta_{i'i} d_i^1 \delta_{jj'} d_j^2$.

We have written the full $(n \times n) \times (n \times n)$ Hessian (i.e. the Hessian restricted to the tangent space of $S_X$). But by the above discussion, we know that this matrix is not invertible (exercise: verify this directly from the above formula!). As in the previous discussion, we need to restrict the matrix to the tangent space of the solution manifold $S'_X$, which can locally be parameterized by the indices $i \leq j$ (and correspondingly $i' \leq j'$). That is, if we have specified the values of $W_{ij}$ for $i \leq j$, then the rest of the values of the matrix are determined by the orthogonal rows constraint. Note that this is a total of $(n+1)n/2$ independent parameters.

Now, consider the $\delta_{ji=i'j'}$ term. Suppose this is non-zero. By the constraints on $i, j$, this implies that
$$i' = j \geq i = j' \geq i',$$
which forces $j = i = i' = j'$.

Therefore, we have
$$H_{ij,i'j'} = (\delta_{ij=i'j'} + \delta_{iji'j'})(\sqrt{d_i/d_j} + \sqrt{d_j/d_i}) \tag{77}$$
with $i \leq j, i' \leq j'$.

If we order the rows by $j$, and then by $i$ (and similarly for the columns), we see that the matrix is block-diagonal, where each block itself is diagonal of the form
$$\sqrt{d_1/d_j} + \sqrt{d_j/d_1}, \ldots, \sqrt{d_{j-1}/d_j} + \sqrt{d_j/d_{j-1}}, 4, \tag{78}$$
the 4 corresponding to the $i = j$ term. So the determinant of the matrix is the product of determinant of the blocks:
$$det_{i \leq j, i' \leq j'}(H_{ij,i'j'}) = 4^n \prod_{i<j\leq n} \sqrt{d_i/d_j} + \sqrt{d_j/d_i}. \tag{79}$$

One claim that we still need to verify is that the tangent space $S'_X$ is the *maximal* subspace on which the Hessian is invertible. But this is easy to see from the above discussion. Indeed, consider the ordering from above in which $H$ is diagonal.

Assume to the contrary that there exist some vector $v$ orthogonal to the tangent vectors $\partial W_{ij}, i \leq j$, such that $Hv = cv$ for some $c \neq 0$. We can write $v$ as a linear combination of the remaining vectors $W_{ij}, i > j$. Therefore, there must exist at least one pair $i_0, j_0$ such that $i_0 > j_0$ and the augmented matrix $\tilde{H}$ given by adding the row and column corresponding to $i_0, j_0$ is invertible.

Using Equation 76 for $H$, it follows that the column $\tilde{H}_{i_0 j_0,:}$ is proportional to the column $\tilde{H}_{j_0 i_0,:}$. This contradiction establishes the claim.

## F.2 Jacobian Computation

To evaluate $J$, we first need to write out $Jac$, or equivalently the derivatives $\partial X_{ij}/\partial U_{ab}$ and $\partial X_{ij}/\partial V_{ab}$ where $X = UV$. A straightforward calculation gives:
$$\partial X_{ij}/\partial U_{ab} = \delta_{ai} V_{bj} \tag{80}$$
$$\partial X_{ij}/\partial V_{ab} = \delta_{bj} U_{ia}. \tag{81}$$

Thus the entries of the matrix $G := Jac(Jac)^T$ are
$$G_{ij,i'j'} = \sum_{ab} \delta_{ai} V_{bj} \delta_{ai'} V_{bj'} + \delta_{bj} U_{ia} \delta_{bj'} U_{i'a} = \delta_{ii'} (V^T V)_{jj'} + \delta_{jj'} (U^T U)_{i'i}.$$

Now specialize to $U = U_{min}$ and $V = V_{min}$, which, recall, by our choice of coordinates, are simply $U_{min} = V_{min} = X^{1/2}$. Thus the matrix simplifies to
$$G_{ij,i'j'} = \delta_{ii'} \delta_{jj'} d_j + \delta_{ii'} \delta_{jj'} d_i = \delta_{ij=i'j'}(d_j + d_i).$$

The matrix is diagonal, so
$$J(U_{min}, V_{min}) = \det(G)^{1/2} = \prod_{ij}(d_i + d_j)^{1/2}.$$

### F.3 VOLUME FORM COMPUTATION

We also need to work out the volume form on $S_X$. We regard $S_X$ as a submanifold of $\mathbb{R}^{n \times n} \times \mathbb{R}^{n \times n}$. We can locally parametrize this by coordinates similarly to the Hessian computation, by $W \in \mathbb{R}^{n \times n}$ and define $U(W) = W, V(W) = W^{-1}X$, which is well defined for $W$ in a sufficiently small neighborhood of $I$.

We need the partial derivatives:

$$\partial U_{ab}/\partial W_{ij} = \delta_{ab=ij} \tag{82}$$
$$\partial V_{ab}/\partial W_{ij} = -(W^{-1}e_{ij}W^{-1}X)_{ab} \tag{83}$$

where $e_{ij}$ is the matrix which is all zeros except for a 1 in the $ij$ entry. By general Riemannian geomtry (cf. Chavel (2006)), the volume form is given by the square root of the determinant of following matrix:

$$G_{ij,i'j'} = \sum_{ab} \partial U_{ab}/\partial W_{ij}U_{ab}/\partial W_{i'j'} + \partial V_{ab}/\partial Z_{ij}W_{ab}/\partial W_{i'j'} \tag{84}$$

$$= \delta_{ij=i'j'} + \sum_{ab}(W^{-1}e_{ij}W^{-1}X)_{ab}(W^{-1}e_{i'j'}W^{-1}X)_{ab}. \tag{85}$$

(Note that this is a different matrix than the matrix $G$ from the preceding section, although both matrices are Gram matrices so we will keep the $G$ notation). The volume form is, as normal, given by square root of determinant of this matrix.

Recall that in our choice of coordinates, the minimum occurs when $W = X^{1/2}$ (and thus $W^{-1}X = X^{1/2}$), so

$$G_{ij,i'j'} = \delta_{ij=i'j'} + \sum_{ab} d_i^{-1/2}\delta_{ia}d_j^{1/2}\delta_{jb} + d_{i'}^{-1/2}\delta_{i'a}d_{j'}^{1/2}\delta_{j'b} \tag{86}$$

$$= \delta_{ij=i'j'} + d_i^{-1/2}d_j^{1/2} + d_{i'}^{-1/2}d_{j'}^{1/2}, \tag{87}$$

using the fact: $(Be_{ij}A)_{ab} = B_{ai}A_{jb}$.

Now, define the vector $v_{ij} := \sqrt{d_j/d_i}$ and let $\mathbf{1}$ be the vector of all ones (of length $n^2$). We can then write the matrix as follows:

$$G_{ij,i'j'} = \delta_{ij=i'j'} + v_{ij}\mathbf{1}^T + \mathbf{1}v_{i'j'}^T \tag{88}$$

$$= \delta_{ij=i'j'} + (v\mathbf{1}^t + \mathbf{1}v^t)_{ij,i'j'} \tag{89}$$

$$G = I + (\begin{array}{cc} v & \mathbf{1} \end{array})(\begin{array}{cc} \mathbf{1} & v \end{array})^T. \tag{90}$$

By Sylvester's determinant identity ($\det(I + AB) = \det(I + BA)$):

$$\det(G) = \det(I + (\begin{array}{cc} \mathbf{1} & v \end{array})^T(\begin{array}{cc} v & \mathbf{1} \end{array})) \tag{91}$$

$$= det\begin{pmatrix} 1 + \sum_{ij} v_{ij} & N^2 \\ \sum_{ij} v_{ij}^2 & 1 + \sum_{ij} v_{ij} \end{pmatrix} \tag{92}$$

$$= (1 + \sum_{ij} v_{ij})^2 - N^2 \sum_{ij} v_{ij}^2 \tag{93}$$

$$= (1 + \sum_i d_i^{-1/2} \sum_j d_i^{1/2})^2 - n^2 \sum_j d_j \sum_i d_i^{-1}. \tag{94}$$

### F.4 COMPLETE RESULT

Plugging all the intermediate results from the preceding subsections back into Equation 57 yields:

$$P(X/\tau) \sim D(n)\frac{1}{\sqrt{\prod_{i<j\leq n}\sqrt{d_i/d_j} + \sqrt{d_j/d_i}}}\sqrt{\frac{(1 + \sum_i d_i^{-1/2}\sum_i d_i^{1/2})^2 - n^2\sum_i d_i\sum_i d_i^{-1}}{\prod_{ij}d_i + d_j}}e^{-\|X\|_*/\tau}, \tag{95}$$

with:

$$D(N) = \frac{n!}{(2\pi\tau)^{3n^2/4 - n/4}} \, . \tag{96}$$

If we just consider the dominant exponential part (i.e. the factor the depends on $\tau$), then we get the marginally less precise formula:

$$P(X/\tau) \quad \sim \quad C\tau^{-3n^2/4 + n/4} e^{-\|X\|_*/\tau} \, , \tag{97}$$

as given in the main text.

We expect the non-square case to be amenable to a similar analysis but leave it for future work.

## G  SPECTRUM OF NORMAL PRODUCT DISTRIBUTION

Here we state some known theoretical results about the Normal Product distribution.

It is easily seen that, just like the Nuclear Norm distribution, the Normal product distribution is symmetric under left and right multipication by orthogonal matrices. Therefore, in a sense, all of the interesting information about this distribution is contained in its spectrum.

In the limit where $n \to \infty$, there are exact analytical formulas for both the eigenvalue density and singular value density for the Normal product distribution Burda et al. (2010a;b). For simplicity, we will state the results for the square case only, although the results for singular values can be extended to the non-square case as well (obviously the matrix must be square to even make sense of eigenvalues in the first place).

In both cases, we will consider a random matrix $Y_n := n^{-1} X_1 X_2$ where $X_1, X_2$ have iid unit normal entries, and take $n \to \infty$.

We begin with the eigenvalue result. In this case, the limiting eigenvalue density $\rho$ in the complex plane is given by Burda et al. (2010a):

$$\rho(z) = \frac{1}{\pi} \mathbf{1}_{|z| \le 1} \frac{1}{|z|} \, . \tag{98}$$

In the singular value case, we consider the limiting density $\rho_{sv}$ of the *squared* singular values. This density satisfies the following cubic equation:

$$\lambda^2 (\pi\rho_{sv}(\lambda))^3 - \lambda(\pi\rho_{sv}(\lambda)) - 1 = 0 \, , \tag{99}$$

where now $\lambda$ is real and positiveBurda et al. (2010b). It is possible to back out an explicit formula for $\rho_{sv}$ using the standard cubic formula.

## H  BAYESIAN ANALYSIS

### H.1  THE GAUSSIAN CASE

$$P(\mathbf{X}|\lambda, \sigma^2) = C(M, N) \left( \frac{\lambda}{\sqrt{\sigma^2}} \right)^{MN} e^{-\frac{\lambda}{\sqrt{\sigma^2}} \|\mathbf{X}\|_*} \tag{100}$$

$$P(\lambda|a_\lambda, b_\lambda) = C(a_\lambda, b_\lambda) \lambda^{a_\lambda - 1} e^{-b_\lambda \lambda} \tag{101}$$

$$P(\sigma^2|a_\sigma, b_\sigma) = C(a_\sigma, b_\sigma) (\sigma^2)^{-a_\sigma - 1} e^{-\frac{b_\sigma}{\sigma^2}} \tag{102}$$

i.e. we have a (rate-parameterized) Gamma prior on $\lambda$ and an inverse-gamma prior on $\sigma^2$, parameterized by the rate of the underlying gamma distribution.

For the $\lambda$ update, we obtain the following full conditional by combining the $\lambda$ prior and $\mathbf{X}$ prior:

$$P(\lambda|\mathbf{X}, \sigma^2) \propto \lambda^{a_\lambda + NM - 1} e^{-\left( b + \frac{\|\mathbf{X}\|_*}{\sqrt{\sigma^2}} \right)\lambda} \, , \tag{103}$$

which is a $\Gamma\left(a + NM, b + \frac{\|\mathbf{X}\|_*}{\sqrt{\sigma^2}}\right)$ distribution.

The $\sigma$ update must include the contribution from the $\sigma$ prior, from the $\mathbf{X}$ prior, and from the likelihood. This yields the following conditional density:

$$(\sigma^2)^{-a_\sigma - \frac{MN}{2} - \frac{|Y|}{2} - 1} e^{-\frac{\left(b_\sigma + \|\mathbf{X} - \frac{1}{2}\mathbf{Y}\|_2^2\right)}{\sigma^2}} e^{-\frac{-\lambda\|\mathbf{X}\|_*}{\sqrt{\sigma^2}}}, \tag{104}$$

where $|Y|$ gives the number of entries observed in $\mathbf{Y}$ (which is $MN$ in the non-completion case).

In the Bayesian Lasso, closed form Gibbs updated are available for $\sigma^2$ by exploiting a mixture representation of the Laplace which seems not to be available in the matrix setting. Therefore, we are left with that final exponential term which complicates the density. We therefore simply use a Metropolis-Hastings step with adaptive proposal variance.

It seems like there is very high correlation between $\lambda$ and $\sigma^2$. It would be nice to sample them jointly. Here is their joint conditional:

$$(\sigma^2)^{-a_\sigma - \frac{MN}{2} - \frac{|Y|}{2} - 1} \lambda^{MN + a_\lambda - 1} e^{-\frac{\|\mathbf{X} - \mathbf{Y}\|_2^2}{2\sigma^2}} e^{-\frac{\lambda}{\sqrt{\sigma^2}}\|\mathbf{X}\|_*} e^{-\frac{b_\sigma}{\sigma^2} - b_\lambda \lambda} \tag{105}$$

Let's sample from the marginal for $\sigma^2$ with $\lambda$ integrated out. That's given as:

$$P(\sigma^2|\mathbf{X}, \mathbf{Y}) \propto \left((\sigma^2)^{-a_\sigma - \frac{MN}{2} - \frac{|Y|}{2} - 1} e^{-\frac{b_\sigma}{\sigma^2}} e^{-\frac{\|\mathbf{X} - \mathbf{Y}\|_2^2}{2\sigma^2}}\right) \int_{\mathbb{R}_+} \lambda^{MN + a_\lambda - 1} e^{-\lambda\left(\frac{\|\mathbf{X}\|_*}{\sqrt{\sigma^2}} + b_\lambda\right)} d\lambda$$

$$\tag{106}$$

$$= (\text{''}) \frac{\Gamma(MN + a_\lambda)}{\left(\frac{\|\mathbf{X}\|_*}{\sqrt{\sigma^2}} + b_\lambda\right)^{MN + a_\lambda}} \propto \left((\sigma^2)^{-a_\sigma - \frac{MN}{2} - \frac{|Y|}{2} - 1}\right) \left(e^{-\frac{b_\sigma + \frac{1}{2}\|\mathbf{X} - \mathbf{Y}\|_2^2}{\sigma^2}}\right) \left(\frac{\|\mathbf{X}\|_*}{\sqrt{\sigma^2}} + b_\lambda\right)^{-(MN + a_\lambda)}$$

$$\tag{107}$$

Then, we can sample $\lambda$ from that Gamma distribution.

## H.2 Unimodality of the Posterior for Gaussian Likelihood

Here we show that if $P_{\gamma^2}(\frac{1}{a^2})$ is log-concave in $a$, $\lambda$ is a fixed constant greater than zero and $Y$ is given by $X$ perturbed additively by iid normal errors, then the joint posterior on $X, \gamma^2$ is unimodal (in the sense of having connected upper level sets) when using the conditional prior $P(X|\gamma^2, \lambda) = \text{NND}(\frac{\lambda}{\sqrt{\gamma^2}})$. The proof strategy, based on that given by Park & Casella (2008) for the Bayesian lasso case, is to establish concavity of the log-posterior under a homeomorphism of the parameters. Since concavity implies that the upper level sets of a function are connected, the homeomorphism preserves the connectedness and we have that the upper level sets are connected in the original parameter space as well.

The log likelihood with respect to $X, \gamma^2$ is:

$$\log P(Y|X, \gamma^2) = C_1 - \frac{MN}{2}\log\gamma^2 - \frac{1}{2\gamma^2}\|Y - X\|_F^2, \tag{108}$$

where $C_1$ is a constant independent of $X$ and $\gamma^2$ and the log prior density of $X|\lambda, \gamma^2$ is :

$$\log P(X|\lambda, \gamma^2) = C_2 - \frac{MN}{2}\log\gamma^2 - \frac{\lambda}{\sqrt{\gamma^2}}\|X\|_*, \tag{109}$$

yielding a posterior density for $X, \gamma^2$ of

$$\log P_{\gamma^2}(X, \gamma^2|Y, \lambda) = C_3 + \log P(\gamma^2) - \frac{\lambda}{\sqrt{\gamma^2}}\|X\|_* - MN\log\gamma^2 - \frac{1}{2\gamma^2}\|Y - X\|_F^2. \tag{110}$$

We now reparameterize as $\Phi = \frac{X}{\sqrt{\gamma^2}}, \rho = \frac{1}{\sqrt{\gamma^2}}$, yielding:

$$\log P(X, \gamma^2|Y, \lambda) = C_3 + \log P_{\gamma^2}(\frac{1}{\rho^2}) - \lambda\|\Phi\|_* + 2MN\log\rho - \frac{1}{2}\|\rho Y - \Phi\|_F^2. \tag{111}$$

Note that we do not need any Jacobian terms here because we are not trying to find the posterior distribution of $\Phi, \rho$. Rather, we are simply using this transformation to analyze the posterior of $X, \gamma^2$.

Since $\|\Phi\|_*$ is convex, its negation is concave, as is the logarithm function. By assumption, $\log P_{\gamma^2}(\frac{1}{\rho^2})$ is also concave. Since concavity is preserved under summation then we need only show that the quadratic term is concave. We establish this by considering the vectorized form of the problem, where $\text{vec} : \mathbb{R}^{m \times n} \to \mathbb{R}^{mn}$ stacks columns of a matrix into a vector:

$$\|\rho Y - \Phi\|_F^2 = \|\rho \text{vec}(Y) - \text{vec}(\Phi)\|_2^2.$$

This norm may be expressed as the following quadratic form:

$$\begin{bmatrix} \text{vec}(\Phi)^\top & \rho \end{bmatrix} \begin{bmatrix} \mathbf{I}_{mn \times mn} & -\text{vec}(Y) \\ -\text{vec}(Y)^\top & \|Y\|_F^2 \end{bmatrix} \begin{bmatrix} \text{vec}(\Phi) \\ \rho \end{bmatrix}$$

$$= \begin{bmatrix} \text{vec}(\Phi)^\top & \rho \end{bmatrix} \left( \begin{bmatrix} \mathbf{I}_{mn \times mn} \\ -\text{vec}(Y)^\top \end{bmatrix} \begin{bmatrix} \mathbf{I}_{mn \times mn} & -\text{vec}(Y) \end{bmatrix} \right) \begin{bmatrix} \text{vec}(\Phi) \\ \rho \end{bmatrix}$$

which exhibits the quadratic coefficient matrix as an outer product and thus as a positive semidefinite matrix, and consequently makes clear that this term will be concave when multiplied by $-\frac{1}{2}$, establishing our result.

### H.2.1 POSTERIOR MULTIMODALITY UNDER THE UNCONDITIONAL PRIOR

In the event that we use the unconditional prior $\log P(X|\lambda) = -\lambda\|X\|_*$, it is easy to encounter multimodality. We created Figure 2 of the main text via the following simple experiment. A "true" $\tilde{X} \in \mathbb{R}^{m \times n}$ was created from iid standard normal entries, and subsequently $Y$ was created by adding iid Gaussian noise with a true variance of $\tilde{\gamma}^2 = 0.01$. We set $\lambda = 15$ and evaluate the posterior density under both the conditional and unconditional prior varying only the first singular value of $\tilde{X}$, denoted $\sigma_1(\tilde{X})$. For $\gamma^2$ we use the reference prior $P(\gamma^2) \propto \frac{1}{\gamma^2}$. That is to say, the 2D function being visualized is:

$$f(a, \gamma^2) = P\left(X = a\mathbf{u}_1\mathbf{v}_1^\top + \sum_{j=2}^3 \sigma_i \mathbf{u}_i\mathbf{v}_i^\top, \gamma^2 | Y, \lambda\right),$$

where $\mathbf{u}, \mathbf{v}, \sigma$ are given by the singular value decomposition of $\tilde{X}$. In the left panel of Figure 2, the mode on the left occurs near $a \approx 0, \gamma^2 \approx \frac{\|Y\|_F^2}{mn}$, which represents the null model, and the mode on the right is near $a \approx \sigma_1(\tilde{X}), \gamma^2 \approx \tilde{\gamma}^2$, which is the "correct" mode insofar as it is closer to the data generating parameters. By contrast, the right panel of Figure 2 shows us that the conditional prior gives us a single mode which we might think of as a compromise between the two modes of the unconditional case.

## I   FURTHER RESULTS ON NORMAL PRODUCT VS. NUCLEAR NORM DISTRIBUTION

In Figure 9 we show the empirical distribution of the Nuclear Norm of the samples, for various matrix sizes. The theoretical Gamma density, as per Proposition 3.2 is superimposed. We can see that the NND samples indeed closely track the Gamma density, although in higher dimensions there are some deviations due to the sampling stochasticity. The Normal Product samples also track the Gamma density reasonably well, but with notable deviations: in particular, their values appear to have a heavier leftward skew than the Gamma.

Figure 10 shows the empirical distribution of the norms of the complex eigenvalues. By results of Burda et al. (2010b) this density should be approximately constant for the Normal Product samples (becoming exactly uniform in the limit of large matrices). We see a directionally similar pattern as in Figure 9, with the Normal Product density placing more mass on smaller eigenvalues compared to the NND.

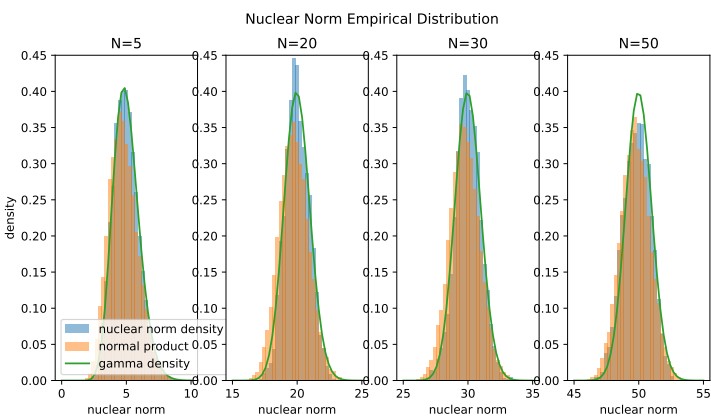

Figure 9: Empirical nuclear norm density, for $\text{NND}(1)$ and $NP(\sigma^2 = 4/3)$ samples.

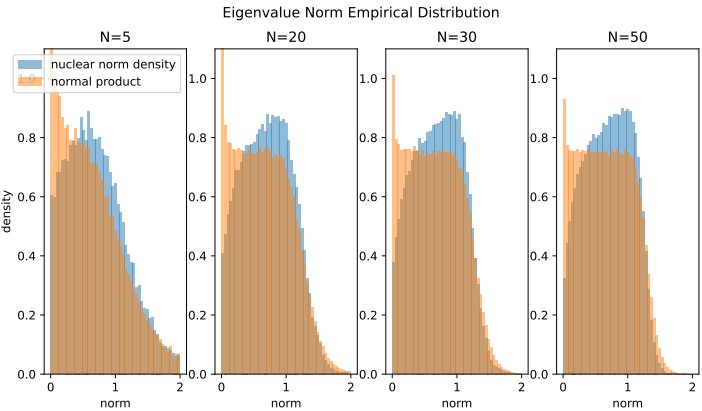

Figure 10: Empirical eigenvalue magnitude density, for $\text{NND}(1)$ and $NP(\sigma^2 = 4/3)$ samples.

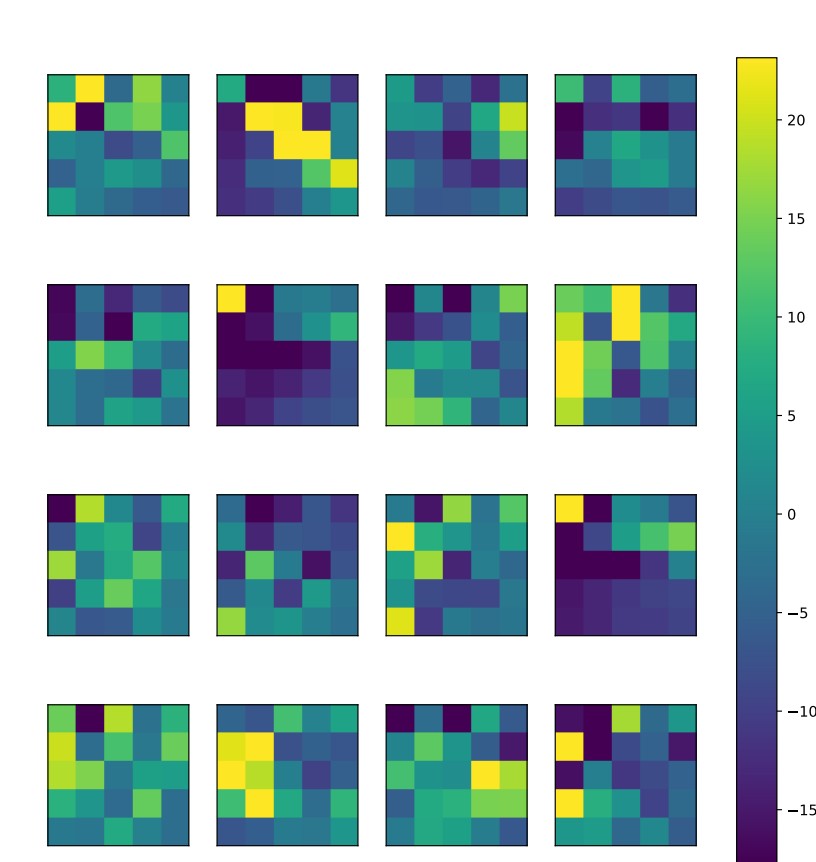

Figure 11: Random samples from the MNIST Fourier transform dataset.

## J    NUMERICAL EXPERIMENT DETAILS

In this section we describe in further detail our numerical results. These were executed in about 8 hours in parallel across nodes of a heterogeneous computing cluster.

### J.1    MNIST DATASET DETAILS

We took the MNIST training dataset and used the `numpy.fft` library to compute the 2-dimensional Fourier transform of each image. We then took the submatrix of the output corresponding to only the first (i.e. lowest frequencies) five rows and columns. We also subtracted the mean of each channel, since the mean of the Nuclear Norm distribution is zero.

Samples from the dataset so obtained are shown in Figure J.1

### J.2    NATURAL IMAGE DATASET DETAILS

We collected images from ten diverse datasets of natural images (see Table 2). We randomly sampled 100 images from each dataset, and resized these images to size $60 \times 60$ using the Pillow library[3].

---

[3] `https://pillow.readthedocs.io/`

| Display Name | Full Name | Source | Brief Description |
|---|---|---|---|
| nature | Texture and Nature Images | Gondur (2024) | Personal photographs of nature. |
| imagenette | Imagenette | Howard (2020); Deng et al. (2009) | Diverse subset of imagenet. |
| satellite | Remote Sensing Image Classification Benchmark | Li et al. (2020) | Satellite imagery. |
| butterfly | Butterfly Image Classification | [4] | Images of 75 butterfly species. |
| crops | Crops Image Classification | Azam (2024) | Crops from Google Images. |
| cityscapes | Cityscapes Image Pairs | Isola et al. (2017) | Segmented Images of Roads |
| sports | 100 Sports Images | [5] | Web images of various sports. |
| cells | Blood Cell Classification | [6] | Microscopy of blood cells. |
| weather | Weather Image Recognition | Xiao (2021) | 11 weather phenomena. |

Table 2: Natural image dataset details.

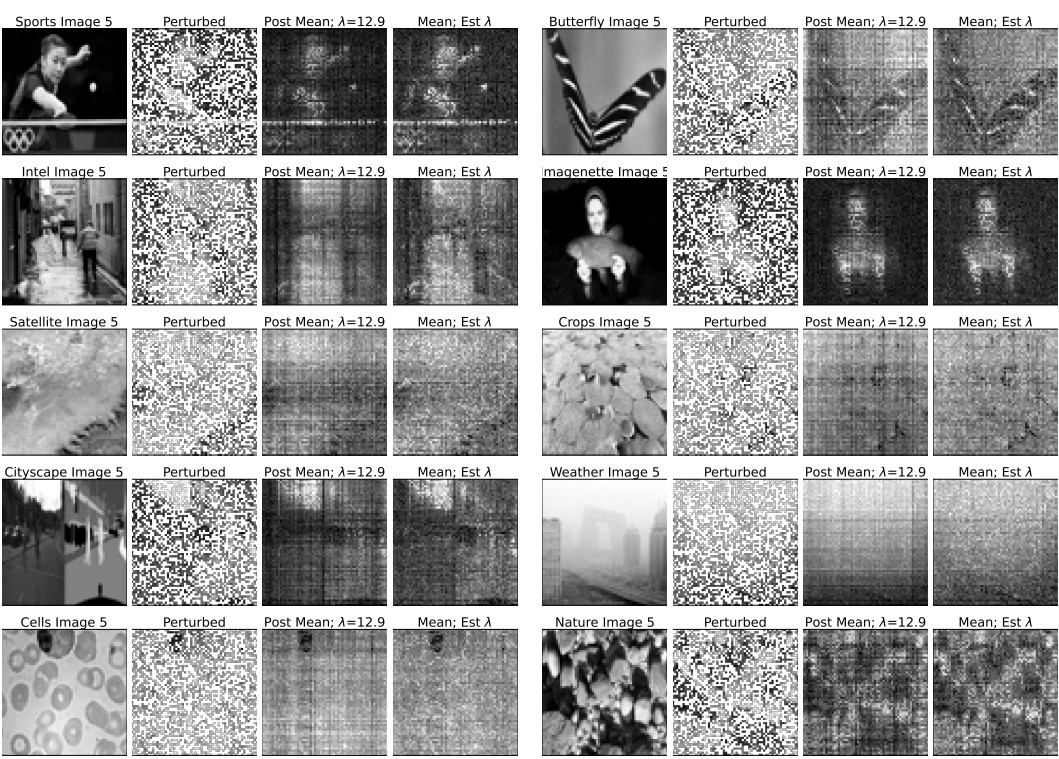

Figure 12: This figure contains an example image from each dataset. Each set of four images gives, from left to right, an original image from each dataset, then the masked and noised array for matrix completion, the posterior mean for nuclear norm method with $\lambda$ fixed to 12.9, and finally the posterior mean of the adaptive method with $\lambda$ estimated.

We extracted the red channel of each image, and then scaled the intensities linearly such that each image had a minimum intensity of 0 and a maximum of 1. Figure 12 shows an example image from each dataset.

### J.3 MATRIX DENOISING

In order to sample from the posterior distribution of a Gaussian likelihood and nuclear norm prior for the matrix denoising problem, we use the Proximal MCMC algorithm of Pereyra (2016); see Appendix A. When using a Gaussian error distribution and the conditional prior, we obtain the

following proximal log-posterior:

$$-\frac{\|Y - X\|_F^2}{2\gamma^2} - \frac{\lambda}{\sqrt{\gamma^2}}\|X\|_* . \tag{112}$$

Therefore, the action of the posterior proximal operator is given by:

$$\text{prox}_s^{-\log P_{X|Y}}(A) = \underset{X \in \mathbb{R}^{M \times N}}{\text{argmin}} \frac{\|A - X\|_F^2}{2s} + \frac{\|Y - X\|_F^2}{2\gamma^2} + \frac{\lambda}{\sqrt{\gamma^2}}\|X\|_* . \tag{113}$$

By completing the square to combine the $F$ norm terms, we are able to apply the standard proximal operator associated the nuclear norm, and ultimately use the following proposal:

$$P(X^*|X^t) = N\left(\text{prox}_{\frac{\lambda\delta\gamma^2}{\delta+2\gamma^2}}^{\|\cdot\|_*}\left(\frac{\delta Y + 2\gamma^2 X^t}{\delta + 2\gamma^2}\right), \delta\mathbf{I}\right) . \tag{114}$$

We adapt $\delta$ as in Section 4.1.

### J.4 Matrix Completion

In the matrix completion case, we have the log-likelihood

$$\log(Y|X, \gamma^2) = -\frac{\|M \odot (X - Y)\|_F^2}{2\gamma^2} \tag{115}$$

where $M \in 0, 1^{M \times N}$ is a masking matrix indicating which pixels are revealed, and $\odot$ denotes elementwise multiplication.

This leads to the following proximal problem:

$$\text{prox}_s^{-\log P_{X|Y}}(A) = \underset{X \in \mathbb{R}^{M \times N}}{\text{argmin}} \frac{\|A - X\|_F^2}{2s} + \frac{\|M \odot (Y - X)\|_F^2}{2\gamma^2} + \frac{\lambda}{\sqrt{\gamma^2}}\|X\|_* . \tag{116}$$

Despite its similarity with the Matrix Denoising case, this proximal operator is not available in a closed form: the $M$ matrix means that the likelihood term is not isotropic, which couples the optimization problems associated with the various singular values and destroys the simple structure allowing for a simple solution. However, as famously demonstrated by the ISTA algorithm, the optimization problem can be solved by iterative application of the nuclear norm proximal operator, which may be viewed as proximal gradient descent. While Pereyra (2016) primarily studies algorithms which use the proximal operator of the entire posterior, they also mention algorithms which use a proximal gradient step as the mean of a Gaussian proposal. This is the approach we take in simulating from the posterior in our matrix completion problems. Here, $\delta$ functions as a gradient descent step size. Again, we adapt it as in Section 4.1.

### J.5 Additional Comparators

|  | MSE | CI |  |
| --- | --- | --- | --- |
| PMF | 0.84 | (0.76, 0.91) | 0.69 |
| SMG | 0.57 | (0.54, 0.60) | 0.94 |
| NP | 0.48 | (0.46, 0.49) | 1.00 |
| NND | 0.55 | (0.54, 0.56) | 1.00 |

Though the focus of this article is to elucidate the probability distribution associated with the nuclear norm, we here briefly provide a preliminary comparison to other existing low-rank inference procedures (see Table J.5) on our matrix denoising benchmark. "PMF" refers to Salakhutdinov & Mnih (2008), using $R = 10$ as is the default in the MATLAB code provided. "SMG" refers to Yuchi et al. (2023), using the default $R = 30$, also a default of the provided MATLAB code. NP and NND are the normal product and nuclear norm distributions studied in this article. The table gives the average performance of the methods across all datasets. The first column gives Mean Squared Error and the second a confidence interval thereon. The final column gives the observed proportion of observations where a method outperformed the naive method of simply predicting the noisy method. Astonishingly, the proposed methods were not once observed doing worse than this naive procedure.

