# OpenReview forum: "A Probabilistic Basis for Low-Rank Matrix Learning"
_ICLR.cc/2026/Conference — Submitted to ICLR 2026_

### Official Review · Reviewer_Dn8r · 2025-10-28

**Soundness:** 2
**Presentation:** 3
**Contribution:** 2
**Rating:** 4
**Confidence:** 2

**Summary:**

This paper approaches conventional matrix completion with nuclear norm regularization from a Bayesian perspective by studying the fundamental distribution $f(X) \propto \exp(-\lambda \|X\|_*)$. Building on this viewpoint, the authors propose an MCMC algorithm tailored for low-rank Bayesian inference and explore strategies for estimating the tuning parameter $\lambda$.

**Strengths:**

This paper seems to be the first to rigorously study the properties of the matrix-valued distribution with density proportional to $ \exp(-\lambda \|X\|_*) $. It presents an explicit formula for the normalizing constant and provides an approximation for the stochastic representation that is both computationally and analytically tractable.

**Weaknesses:**

It appears that the proposed method can assist in estimating the optimal tuning parameter $\lambda$ without relying on a grid search, which is commonly used in practice. While some modifications have been proposed to boost the MCMC algorithm, but it is still relatively slow compared to the well-established soft-impute algorithm. I am curious whether there are additional benefits to adopting a Bayesian approach for matrix learning. Although I am not an expert in Bayesian methods, one notable advantage is the ability to quantify uncertainty through the posterior distribution. In contrast, traditional matrix completion methods often face challenges in conducting inference. It would strengthen the paper if the authors could further illustrate—both theoretically and numerically—the inference capabilities of their estimator for the target matrix.

**Questions:**

Before equation (3), it says the constant $C$ is independent of $X$. I am curious, isn't $C$ will depend on the dimension of $X$?

**Details Of Ethics Concerns:**

N/A.

---

### Official Review · Reviewer_L6sh · 2025-10-30

**Soundness:** 3
**Presentation:** 2
**Contribution:** 3
**Rating:** 6
**Confidence:** 3

**Summary:**

This paper primarily elucidates probability distributions related to the nuclear norm. By studying the distribution with density, the authors develop an improved MCMC algorithm for low-rank Bayesian inference and propose a principled approach to learn the penalty parameter , thereby eliminating the need for hyperparameter tuning. Research indicates that leveraging fundamental properties of the nuclear norm distribution can enhance the accuracy and efficiency of low-rank Bayesian matrix denoising and completion algorithms.
Paper is mostly theory with some lightweight experiments.

**Strengths:**

1. The paper presents rigorous theoretical derivations with strong conceptual significance, offering a novel probabilistic perspective that bridges convex optimization and Bayesian inference. In addition, it systematically investigates the probabilistic foundations underlying the nuclear norm regularization term, providing an explicit analytical form of the nuclear norm distribution, along with its normalizing constant, symmetry properties, and the distribution of singular values.

2. The motivation behind the problem formulation and algorithm design is clearly articulated, and the numerical experiments cover a wide range of scenarios.

**Weaknesses:**

1. The authors claim to have designed an improved MCMC algorithm for low-rank Bayesian inference, as described in Section 4.1 and the corresponding appendix. However, the paper does not clearly specify how the proposed MCMC algorithm differs from existing ones, either theoretically or empirically. It is recommended that Section 4.1 be expanded to include a clearer theoretical analysis and experimental comparison to substantiate the claimed improvement.

2. In Section 5.2, the numerical experiments evaluate the effective sample size (ESS) of the proximal and SVD-Langevin MCMC methods only on synthetic rank-1 data. Conducting additional experiments on non-synthetic benchmark datasets—particularly under varying rank settings—would strengthen the empirical evidence and enhance the overall persuasiveness of the paper.

3. The design purposes of Figures 2 and 3 are unclear and somewhat confusing. In addition, Figure 4 lacks quantitative evaluation metrics. A more detailed analysis of how the parameter  affects the algorithm’s performance, along with a broader set of experiments, would significantly strengthen the credibility and impact of the study.

**Questions:**

1. I would like to see applications of the proposed algorithm in more practical scenarios, such as robust PCA or related real-world low-rank inference tasks. Furthermore, a performance comparison with other representative methods—such as those based on the weighted nuclear norm—would provide a more comprehensive evaluation and highlight the advantages of the proposed approach.

2. How does the computational complexity of the proposed improved MCMC method scale as the matrix dimension increases? Has the convergence rate or computational cost been quantitatively analyzed either theoretically or empirically?

---

### Official Review · Reviewer_NPQk · 2025-11-06

**Soundness:** 3
**Presentation:** 4
**Contribution:** 3
**Rating:** 6
**Confidence:** 2

**Summary:**

This paper investigates the underlying probability distribution associated with the nuclear norm, a widely used regularizer in low-rank matrix learning. Using tools from differential geometry, the authors analyze its key properties, including the normalization constant, the distribution of the nuclear norm itself, and the joint distribution of its singular values, while also revealing its close relationship with the Normal Product (NP) distribution.
Based on these theoretical results, the paper designs an improved MCMC algorithms.Even when tuning the parameters is difficult or impossible, it can adaptively adjust the hyperparameter λ, leading to better performance in Bayesian low-rank matrix denoising and completion experiments.

**Strengths:**

1. This article fills a theoretical gap. Although the nuclear norm is extensively used in optimization, its probabilistic characterization has rarely been studied in depth. The paper provides a clear and valuable theoretical treatment of this topic.

2. The proposed Normal Product (NP) approximation and corresponding sampling procedures bridge theory and practice. The adaptive λ estimation process is well motivated and empirically validated through experiments on matrix completion and denoising tasks.

Overall, this paper presents reliable and meaningful theoretical results, and the experimental section also demonstrates the benefits of this theoretical analysis, such as the estimation of lambda.

**Weaknesses:**

The authors mention that several researchers in variational Bayesian inference have previously noted the close connection between the “Normal Product” distribution and the nuclear-norm distribution. It would strengthen the contribution if the paper could clarify the extent of prior work—both theoretical and applied—by these authors, to better situate the novelty of this study within the existing literature.

**Questions:**

1. Bayesian estimation of the regularization parameter λ is a good subject of study, but it remains unclear how the proposed samplers or adaptive λ estimation scale to high-dimensional matrices. Providing computational complexity indicators, runtime, or empirical timing results would make the work’s practical relevance more convincing.

2. Regarding the approximation of NP to NND, what are the specific assumptions behind this approximation (e.g., the singular value condition, etc.)? Although some experimental results later in the paper show that the two are in agreement, theoretical error analysis or limits (even approximate ones or some discussion) will help to clarify the accuracy and applicability of this approximation.

---

### Official Review · Reviewer_qvqW · 2025-11-06

**Soundness:** 3
**Presentation:** 2
**Contribution:** 3
**Rating:** 4
**Confidence:** 3

**Summary:**

The main result of the paper is the exact computation of the normalizing constant for the nuclear norm distribution. The knowledge of this normalizing constant makes it possible to refine and improve a number of sampling algorithms (prior or posterior based)

**Strengths:**

The paper definitely provides an answer to an interesting question.

**Weaknesses:**

The difficulty I have has to do with the how dense it is. I guess it coud be better organized to fit into the ICLR size limit. But at this stage I feel like the authors just rapidly extracted part of their important work to fit into the first 13 pages and put everything else into the appendix. The paper is good but it should be rewritten around a main clear narrative. In particular I would suggest making a clean introduction to the sampling schemes (with perhaps a quick description along with the definition of the distribution so that it is clear your result is applicable in practice). Then explain when you replace the use of the NND by the NPD. Because in the current version of the paper this is not very clear. You introduce then NPD as a simplification of the NND but when you introduce the sampling algorithms, you go back to the NND and seem to sample directly from the distribution given by Proposition 3.4.

Section 4.1. which should be central to the paper as it deals with the practical use of the nuclear norm distribution appears sketchy and some of the Figures (I think of Figure 6 in particular are ambiguous). To be fair the authors should also be more straightforward regarding the fact that sampling can be done without knowing the normalizing constant of the distribution (this only appears in the numerical experiments if I’m not wrong the authors indicate that some earlier work by Pereyra conduct simulations from the NND without the need for this constant). You also really need to better format your references. Some of them appear in the middle of the text.

====================================================================================


- Proposition 3.2. I would detail a bit more. In particular, I guess the \lambda^{-nm} comes from the determinant of the Jacobian. I would add a word
- Proposition 3.3. I would recall the definition of the surface measure. Also to use the surface measure, don’t you need some regularity assumptions to define the surface measure. How do you know that the set {X|\|X\|_*\leq 1} is sufficiently regular?
- In proposition 3.4. If X = USV^T and U, V and S are defined as in Proposition 3.4. How do you ensure the dimensions match? To me, unless there is something I don’t see, U is of size n \times n, V is of size m\times but then S I assume is of size n\times n ? In this case,
- I would recall the definition of the Stiefel manifold
- In the proof of Proposition 3.4. you mention the book of Zeitouni, Anderson and Guionnet. First you should correct the reference. Anderson and Zeitouni are two different researchers. Second, when you cite the book, it does not make sense to refer to Anderson alone (there should at least be an “et al.“) but this might be related to the way you encoded the reference. Finally [[Check Proposition 4.1.3.]]
- lines 180-181, I don’t understand why you say that it is undefined on the set of measure 0 corresponding to repeated singular values. If I look at the statement of Proposition 3.4. I understand that the probability of having a singular value with multiplicity > 1 is zero (although I don’t really understand why) but I don’t see why the pdf would be undefined on matrices with repeated singular values.
- line 171, what do you mean by “an order constraint” on the singular values ? you mean you assume the singular values are sorted in your notations?
- In Figure 1, the motivation for the display of the Gamma(7,1) diagram is unclear
- line 189, Theorem 3.4. —> Proposition 3.4. ?
- The whole paragraph 182-189 should be rewritten. I would for example remove the comment on the Pereyra paper. This paper seems to be on a completely different topic
- line 226 I would remove the sentence “This is well known Gaunt (2022)” by the simpler “X_1X_2 has the following density (see for example Gaunt 2022)”
- Can you provide a reference for lines 230-234?
- The equality in (11) (i.e. rightmost) is not clear to me. I understand you get something that is proportional to $\frac{1}{\sqrt{z}}e^{-Cz}$ but why do you get rid of the constants (in particular the one in the exponential) ?
- “of the modified Bessel function of second kind Johnsson & Kotz..” —> “of the modified Bessel function of second kind (see e.g. Johnsson & Kotz)”?
- line 261, is that the function on line 1039 ? Then why not include it in the statement of the Theorem?
- Line 264-265 : “it is necessary to restrict the dimensionality” —> What do you mean? I would either remove or give a clearer explanation. Lines 267 to 272 are interesting as they motivate the 4/3 factor but you need to better explain where the 3/4 comes from.
- Sometimes you use NP sometimes you use NPD for the nuclear product (distribution) prior. I recommend homogenizing.

Section 4.1

Generally speaking, there is too much information in section 4.1. and 4.2

- line 292 “relies on the smoothness of the posterior, structure unavailable ..” --> the sentence does not make sense to me.
- line 295 : “and then proposing..” —> do you mean “sampling” ?
- In section 4.1. you use many lines to expand on how how some sampling scheme can or cannot be used on the nuclear norm distribution. Why not just keep the exposition simple and focus on the sampling scheme that are best. Then give the steps associated to those clearly. You should clearly recall what you mean by posterior and prior sampling. Do you mean (from what I understand from line 309-310) that you have Y = F(X) and you are interested in P(X|Y) \propto p(Y|X)p(X) where p(X) \sim NND? I think you should clarify that somewhere. From the same lines, sampling from the prior is then X \sim NND(\lambda)

Section 4.2.

- In the statement of Theorem 4.1. I guess you mean Y|X  \sim N(X, \gamma^2 I) ?
- What is the point of taking gamma random ? I think you should specify P_{\gamma^2} in section 4.2.
- line 322 : “We now consider how a Gibbs sampler can be constructed for this posterior” —> which posterior ? P(X, \gamma^2|Y, \lambda) ? I would then clearly say “from the posterior P(X, \gamma^2|Y,..)”
- There is a mistake in the second appearance of the posterior P(X, \gamm^2|Y) in the statement of Theorem 4.1. If I’m not mistaken, it should be P(X, \gamma^2|Y, \lambda) and not P(X, \gamma^2|X, \lambda)
- I don’t understand how you go from the distributions in Theorem 3.4. to those on lines 324 - 326

**Questions:**

see above

---

### Author Response · Authors · 2025-12-02
**Summary of Changes**

Dear reviewers: thanks so much for your very helpful comments. We have made significant changes to the manuscript in response to your feedback. We regret that this time spent making changes ultimately meant that, due to the OpenReview leak, we did not have time to discuss whether they matched your vision for our article and further iterate, but they have doubtless lead to an improved manuscript nevertheless.

For the convenience of the AC, we will first A) list the major changes to our study, and subsequently B) outline how the changes correspond to suggestions made by each reviewer.

A) Main Substantive Changes:

1) **Larger test images:** previously, we had been downscaling images in our datasets to be about 100x100. We no longer do so, using the native image sizes, with dimensions up to 600.
2) **Improved Results Presentation**: previously, we tried to present in a single figure both 1) the ability of the proposed Bayesian methods to automatically select the regularization strength and 2) the relative performance of qualitatively different approaches, leading to confusion. We have now produced a single table (on Page 8) giving comparative results, and use separate figures to illustrate the adaptation.
3) **Uncertainty Quantification** We now use the Interval Score to assess the probabilistic predictions made by Bayesian models rather than only comparing MSE.
4) **Rank as a Hyperparameter**: We now include different ranks for competing methods rather than just using the default settings.
5) **Emphasize relationship between Normal Product and Nuclear Norm Distributions**: Our novel theory showing how the normal product has asymptotic behavior which exhibits a Nuclear norm sheds new light on the Normal Product approach, in particular showing how the very fact that we are using a product of two normals induces approximate low-rank structure, without the need to explicitly truncate the rank. Our novel numerical experiments bear this out, by demonstrating that full-rank normal product priors can have better performance than recently proposed and much more complex methods. We now communicate this more clearly in the manuscript.

B) How reviewer concerns prompted our changes:

- **Reviewer qvqW**: Thanks for your careful attention to the style and prose of this article which has already lead to improved clarity. Thanks also for calling attention to the confusing presentation of results; we think that the table we now have is much clearer.
- **Reviewer NPQk**: Thanks for probing our theoretical underpinnings and suggesting larger matrices in our experiments. We have increased the matrix size by about an order of magnitude, and observe promising numerical behavior.
- **Reviewer L6sh**: Thanks for pointing out that the presentation of the Bayesian analysis could be clearer; we have better incorporated references to Figures 1 and 2 in the text to clarify their meaning. Also, we have added discussion of computational complexity (dominated by the SVD calculations required for proximal operator computation) to the appendix. Finally, we hope the bigger images also addresses some your concerns.
- **Reviewer Dn8r**: Thanks for your excellent suggestion to include probabilistic forecasting evaluation methods; it ended up showing a whole new dimension along which the methods we propose excel. Additionally, you are exactly correct that the constant in the density depends on the matrix dimensions (but not its contents); we have updated the text to reflect this.

Thanks everyone for your thoughtful comments and we are very grateful for your efforts which have lead to a significantly improved manuscript.

---

### Meta-Review · Area_Chair_ZdmG · 2026-01-14

**Summary:**

**Reviewer qvqW:** Presentation is dense, and organization of the paper can further be improved. Distinguish narrative and parenthetical in-text citations. A long list of points requiring revision is provided.

**Reviewer NPQk:** Unclear novelty. Scalability of the samplers of $\lambda$ to high-dimensional matrices is not clear. Accuracy and applicability of approximating NND with NP is not clear.

**Reviewer L6sh:** Unclear difference between the proposed MCMC algorithm and existing ones. Experiments on non-synthetic data are missing. Some figures are unclear and even confusing. Scalability to higher-dimensional matrices is unclear.

**Reviwer Dn8r:** Benefits of the proposal in Bayesian matrix learning are not clear.

**Additional points:**
- Lines 106-107: I think that one should regard the fact that $e^{-\lambda ||X||\_\*}$ is normalizable (and thus yields a valid probability density) to have already been established, and cannot be regarded as an original contribution of this paper. Indeed, the density of $X$ can be convreted to the densities of $U,V,S$ via applying the change-of-variables formula. The densities of $U$ and $V$ are Haar on compact manifolds, and the density of $S$ has already been established as in Proposition 4.1.3 in Anderson et al. (2010), as mentioned in the proof of Proposition 3.4 in this paper. These collectively imply that $e^{-\lambda||X||\_\*}$ yields a valid density.
- Proposition 3.1: One might have to assume that $U$ and $V$ are *fixed* orthogonal matrices (or independence of $X,U,V$).
- Line 122: Jacobian determinant $\pm1$(.)
- Equation (7): "$min$" should be upright "$\min$".
- Figure 2: $\log\sigma^2$ → $\log\gamma^2$; $-\log P(X|\lambda,\sigma^2)$ → $-\log P(X|\lambda,\lambda^2)$; $\frac{\lambda}{\sqrt{\sigma^2}}$ → $\frac{\lambda}{\sqrt{\gamma^2}}$
- Figure 2 caption: x-axis → horizontal axis; y-axis → vertical axis
- Lines 269-271: The explanation for the factor $3/4$ is not really convincing. In equation (12), it is $\tau$ that primarily determines the scale of $X$, and thus it should not be so straightforward to understand the claimed role of the exponent of $\tau$ as the scaling of $X$.
- Lines 296-297: The relation between $P$ and "the distribution from which samples are desired" is not clear.
- Lines 324-328: Why is the derivation of these formulas not shown?
- Figure 4: This figure does not seem to be referenced in the main text.
- Lines 355-358: Why is the derivation of these formulas not shown?;  $X\gamma^2$ → $X, \gamma^2$
- Line 448: The referred figure (Figure 3) was placed on page 6, which I think is too far away from the referring position.
- Line 466: The section title "Matrix denoising" does not seem to correspond to the content of that section, which is Bayesian estimation of $\lambda$.
- Line 481: to (fixed) a grid of fixed $\lambda$.
- Some parts of Appendices were written in a mood akin to a casual lecture note, which I believe is inappropriate as a material such as a conference paper: "refresher" (line 877), "exercise: verify this directly from the above formula!" (lines 1137-1138).
- Lines 1054, 1080: The multiletter symbols $Jac,VF$ may be incorrectly interpreted as products of several quantities and thus should be avoided.
- Equation (65): One may delete $dF(W)$ on the left-hand side.
- Equations (68)-(73): As $H$ is not a matrix but an order-4 tensor, what these formulas represent should be clarified.
- Equations (84), (86):  Enclose the summand with a pair of parenthesis.
- Equation (95): $\prod_{i<j\le n}\sqrt{d_i/d_j}+\sqrt{d_j/d_i}$ → $\prod_{i<j\le n}(\sqrt{d_i/d_j}+\sqrt{d_j/d_i})$; $\prod_{ij}d_i+d_j$ → $\prod_{ij}(d_i+d_j)$
- Equation (96): $D(N)$ → $D(n)$
- Equation (110): $P(\gamma^2)$ → $P_{\gamma^2}(\gamma^2)$
- Line 1587: $M\in 0,1^{M\times N}$ → $M\in\\\{0,1\\\}^{M\times N}$

**Reviewer Concerns:**

- **Scalability (NPQk, L6sh):** In the revision experiments on larger images have been included.
- **Presentation issues (NPQk, L6sh):** Figure 4 in the original manuscript has been converted to Table 1 in the revised manuscript to summarize the experimental results. I think that several other presentation issues still remain.

**Reviewer Scores:**

The initial evaluations of Reviewers L6sh, NPQk were positive. I would second Reviewer NPQk in that there should be several points to be addressed in this paper (see the list of mine shown above).

---

### Decision · Program_Chairs · 2026-01-26

Reject